# `softmax` IS NOT ENOUGH
# (FOR SHARP OUT-OF-DISTRIBUTION)

## ABSTRACT

A key property of reasoning systems is the ability to make *sharp* decisions on their input data. For contemporary AI systems, a key carrier of sharp behaviour is the `softmax` function, with its capability to perform differentiable query-key lookups. It is a common belief that the predictive power of networks leveraging `softmax` arises from "circuits" which sharply perform certain kinds of computations consistently across many diverse inputs. However, for these circuits to be robust, they would need to generalise well to *arbitrary* valid inputs. In this paper, we dispel this myth: even for tasks as simple as finding the maximum key, any learned circuitry *must disperse* as the number of items grows at test time. We attribute this to a fundamental limitation of the `softmax` function to robustly approximate sharp functions, prove this phenomenon theoretically, and propose *adaptive temperature* as an ad-hoc technique for improving the sharpness of `softmax` at inference time.

## 1 MOTIVATION

It is no understatement to say that the $\mathtt{softmax}_\theta : \mathbb{R}^n \to [0,1]^n$ function[1]:

$$\mathtt{softmax}_\theta(\mathbf{e}) = \begin{bmatrix} \dfrac{\exp(e_1/\theta)}{\sum_k \exp(e_k/\theta)} & \cdots & \dfrac{\exp(e_n/\theta)}{\sum_k \exp(e_k/\theta)} \end{bmatrix} \tag{1}$$

is one of the most fundamental functions in contemporary artificial intelligence systems.

The role of `softmax` in deep learning is to convert any vector of *logits*, $\mathbf{e} \in \mathbb{R}^n$, into a *probability distribution*, in a form that is part of the *exponential* family. Further, `softmax` allows for application of a *temperature* parameter, $\theta \in \mathbb{R}$, to adjust the amount of probability mass attached to the highest logit—a concept borrowed from the Boltzmann distribution in statistical mechanics.

Initially, the primary utilisation of `softmax` in deep learning was within the final layer of *classifiers*. Its influence in this domain vastly expanded after it saw use in the *internal* layers—as a differentiable key-value store (Graves et al., 2014) or a mechanism for *attending* over the most relevant parts of the input (Bahdanau et al., 2015). This *attentional* framing of `softmax` was critical in defining important models for sequences (Vaswani et al., 2017, Transformers), images (Dosovitskiy et al., 2021, ViTs) and graphs (Veličković et al., 2018, GATs).

Several efforts attribute the success of `softmax` to its capability of modelling computations relevant to reasoning. This can be related to the concept of *circuits* in theoretical computer science (Arora & Barak, 2009). Several interpretable pieces of "circuitry" (Olah et al., 2020) have already been discovered in large Transformers, primarily under the umbrella of *mechanistic interpretability* (Elhage et al., 2021; Olsson et al., 2022; Wang et al., 2022).

Here we study the robustness of such circuitry, especially when going beyond the distribution the models are trained on—a critical regime for *reasoning engines*. We find that, in spite of its many successes, `softmax` *does not have a chance* to robustly generalise such circuits out of distribution, especially as it provably cannot approximate **sharpness** with increasing problem size (Figure 1).

Here we call a function taking a variable number of inputs *sharp* if its output value can be expressed using only a *constant* number of these inputs. For example, max is sharp, as its output value is equal

---

[1]Strictly speaking, the proper name for this function should be `soft`**arg**`max`. We choose to retain the terminology introduced by Bridle (1989), primarily for reasons of alignment with modern deep learning frameworks.

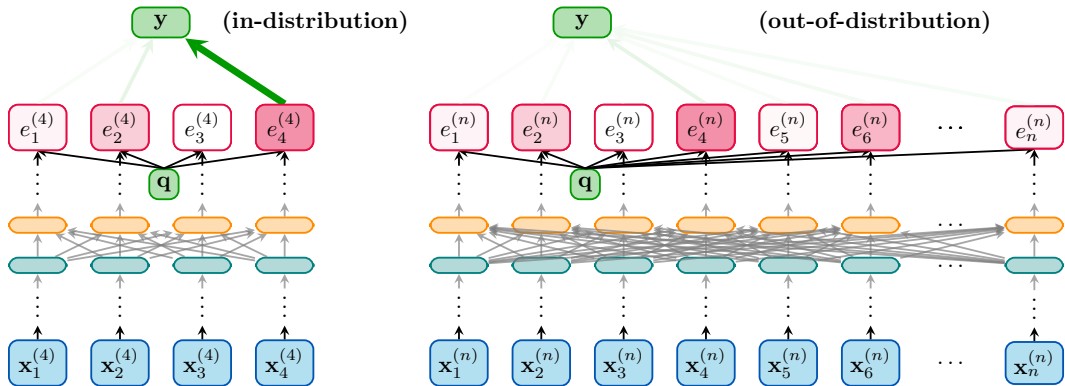

Figure 1: Illustration of Theorem 2.2, one of our key results. Assuming a tokenised input from a fixed vocabulary and a non-zero temperature, for every `softmax` attention head inside an architecture comprising only **MLPs** and **`softmax` self-attention layers**, it must hold that, given sufficiently many **tokens**, its **attention coefficients** will *disperse*, even if they were sharp for in-distribution instances.

to exactly one of its inputs' values. The average function is not sharp, as its output value depends on all of its input values (with factor $1/n$ for each of the $n$ items).

**Key theoretical result**   We define sharp functions by their behaviour as their number of inputs varies. This directly motivates the *out-of-distribution* setting we study: generalising to different amounts of inputs. Specifically, when we analyse neural networks that learn sharp functions, we assume that they are trained on problem instances containing no more than $n$ input items, and we take a particular interest in their sharpness on instances with $n' > n$ items; these are considered *out-of-distribution* instances because they go beyond the maximal number of inputs the model had been prepared for. In language modelling, this setting is also known as *length generalisation* (Anil et al., 2022); in graph machine learning, it is known as *size generalisation* (Yehudai et al., 2021).

Through one of our key theoretical results (Theorem 2.2), we demonstrate that modern deep learning architectures, operating over a fixed vocabulary of input tokens and leveraging the `softmax` function, are fundamentally incapable of learning functions that remain sharp under such out-of-distribution instances. This is due to the fact that the coefficients emitted by the `softmax` function must *disperse* as we increase the number of input items. Here by dispersing we mean that, as the number of input items grows, the coefficient attached to each individual item must decay towards zero. This makes it impossible to robustly compute functions that depend on any particular finite amount of input values, such as the aforementioned `max`, as we show in Appendix B (Corollary B.1 and Remark B.2).

We hope that our results will encourage future study of alternative attentional functions, in light of the problems we identify, especially for building reasoning engines of the future. That being said, we also believe our findings indicate ways to modify the `softmax` function to support sharpness for longer—as one simple example, we propose an *adaptive temperature* mechanism for `softmax`.

**Background**   The analysis of attentional coefficients and attempting to attribute interpretable operations to them dates back to the earliest deployments of internal `softmax` layers at scale; examples include (Graves et al., 2014, Figure 6), (Bahdanau et al., 2015, Figure 3), (Vaswani et al., 2017, Figures 3–5) and (Qiu et al., 2018, Figure 5). A strong current in this space analyses the self-attentional heads of Transformers (Voita et al., 2019; Jain & Wallace, 2019).

With the rise of large language models, mechanistic interpretability has taken charge in detecting and elucidating various circuits in Transformers (Elhage et al., 2021). Some prominent discoveries include induction heads (Olsson et al., 2022), indirect object identification (Wang et al., 2022), multiple-choice heads (Lieberum et al., 2023), successor heads (Gould et al., 2023), attentional sinks (Darcet et al., 2023), comparator heads (Hanna et al., 2024) and retrieval heads (Wu et al., 2024). Most recently, these efforts have relied on sparse autoencoders (Kissane et al., 2024).

The skills above are quite impressive and span many rules one might hope a robust reasoning system would have, and the discovered heads always appear sharp when inspected on *in-distribution* samples. However, it is also known that many *easy* tasks requiring sharp attention—such as finding minima—are hard to do reliably with LLMs *out-of-distribution* (Markeeva et al., 2024, Figure 6). More challenging sharp order statistic tasks, such as finding the second minimum (Ong & Veličković, 2022) may even be hard to learn in-distribution. The discrepancy of such results with the previous paragraph motivate our study, and formalisation of `softmax` dispersion.

Certain dispersion effects in `softmax`—e.g. as an effect of increasing temperature—are already well-understood in thermodynamics. A core contribution of our work is understanding dispersion in a setting where the **amount of logits can vary**, which is relevant for generalisation in Transformers. We are not the first to observe dispersion in this setting empirically; prior works studying the capability of Transformers to execute algorithms (Yan et al., 2020) and perform random-access lookups (Ebrahimi et al., 2024) also note dispersion patterns. Our work is the first to rigorously prove these effects, directly attribute them to the `softmax` operator, as well as propose ways to improve sharpness empirically within `softmax`. The proof technique we will use to demonstrate this is inspired by Barbero et al. (2024), though unlike their work, our key results apply regardless of whether the computational graph is bottlenecked or not.

**Primer on attentional heads and Transformers**    Within this paper we will primarily study the use of `softmax` within *self-attentional* neural network architectures, such as Transformers (Vaswani et al., 2017). The core building block of such models is the (dot-product) *attentional head*, which operates over a collection $n$ of nodes (or tokens), with features $\mathbf{x}_i^{(n)} \in \mathbb{R}^k$ for node $1 \leq i \leq n$, for a given query vector $\tilde{\mathbf{q}}^{(n)} \in \mathbb{R}^k$.

First, the attentional head computes *key*, (updated) *query* and *value* vectors via matrix multiplication:

$$\mathbf{k}_i^{(n)} = \mathbf{K}\mathbf{x}_i^{(n)} \qquad \mathbf{q}^{(n)} = \mathbf{Q}\tilde{\mathbf{q}}^{(n)} \qquad \mathbf{v}_i^{(n)} = \mathbf{V}\mathbf{x}_i^{(n)} \qquad (2)$$

where $\mathbf{K}, \mathbf{Q}, \mathbf{V} \in \mathbb{R}^{k' \times k}$ are learnable parameter matrices. Then, dot-products between the query and all of the key vectors are taken to compute unnormalised attentional coefficients of each item, also known as *logits*, $e_i^{(n)} \in \mathbb{R}$. These coefficients are normalised using the `softmax` function to obtain *attentional coefficients*, $\alpha_i^{(n)} \in \mathbb{R}$. Finally, the attentional coefficients are used for a weighted sum of value vectors, which represents the output of the attentional head, $\mathbf{y}^{(n)} \in \mathbb{R}^{k'}$:

$$e_i^{(n)} = \left(\mathbf{q}^{(n)}\right)^\top \mathbf{k}_i^{(n)} \qquad \alpha_i^{(n)} = \mathtt{softmax}_\theta(\mathbf{e}^{(n)})_j \qquad \mathbf{y}^{(n)} = \sum_{1 \leq i \leq n} \alpha_i^{(n)} \mathbf{v}_i^{(n)} \qquad (3)$$

With regard to how attentional heads are used within Transformers, we will mainly analyse two of the most popular strategies: BERT-style (Devlin et al., 2019) and GPT-style (Radford et al., 2018). In both cases, each of the input nodes computes its own attentional output, i.e. there is one query vector per node, computed as $\mathbf{q}_i^{(n)} = \mathbf{Q}\mathbf{x}_i^{(n)}$, leading to per-node attention coefficients $\alpha_{ij}$ and outputs $\mathbf{y}_i^{(n)}$ by distributing Equation 3 across queries. The main difference is in the choice of keys.

In BERT-style self-attention, each node's query vector attends over all of the key vectors, i.e. it is obtained by directly distributing Equation 3 across all queries:

$$e_{ij}^{(n)} = \left(\mathbf{q}_i^{(n)}\right)^\top \mathbf{k}_j^{(n)} \qquad \alpha_{ij}^{(n)} = \mathtt{softmax}_\theta(\mathbf{e}_i^{(n)})_j \qquad \mathbf{y}_i^{(n)} = \sum_{1 \leq j \leq n} \alpha_{ij}^{(n)} \mathbf{v}_j^{(n)} \qquad (4)$$

In comparison, GPT-style attention (also known as "causal masking" or the decoder-only Transformer) only allows information to flow *forwards*; each node's query vector may only attend to the key vectors from nodes that precede it. This yields the following modification:

$$e_{ij}^{(n)} = \begin{cases} \left(\mathbf{q}_i^{(n)}\right)^\top \mathbf{k}_j^{(n)} & j \leq i \\ -\infty & j > i \end{cases} \qquad \alpha_{ij}^{(n)} = \mathtt{softmax}_\theta(\mathbf{e}_i^{(n)})_j \qquad \mathbf{y}_i^{(n)} = \sum_{1 \leq j \leq i} \alpha_{ij}^{(n)} \mathbf{v}_j^{(n)}$$
$$(5)$$

Our key dispersion results hold for both styles of attention—this is mainly due to the fact that all predictions made by GPT-style architectures are dependent on the *final* token embedding, $\mathbf{y}_n^{(n)}$, which will attend over all items, much like any BERT head. The main difference between the two will be in qualitative effects on certain corollaries of the theory (Appendices B–C).

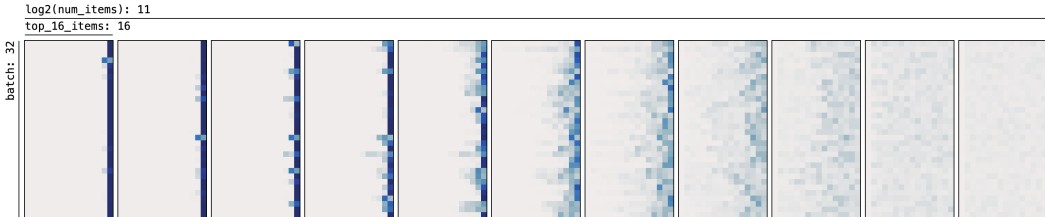

```
log2(num_items): 11
top_16_items: 16
```

Figure 2: Visualising the attentional head for the `max` retrieval task for a batch of 32 randomly-sampled input sets (each represented by one of the rows), over the 16 items with largest key (columns). If the head operates correctly, it must allocate sharp attention to the rightmost item. From left to right, in each frame we *double* the number of items the head has to process.

## 2 DISPERSION IN `softmax` AND TRANSFORMERS

To motivate our theory, we train a simple architecture including a single attentional head to predict a feature of the *maximum* item in a set. Each item's features are processed with a deep MLP before attending, and the output vector of the attention is passed to a deep MLP predictor (see Appendix A for experimental details). We train this model using sets of $\leq 16$ items, and in Figure 2 we visualise the head's attentional coefficients, computed over sets of varying size at inference time.

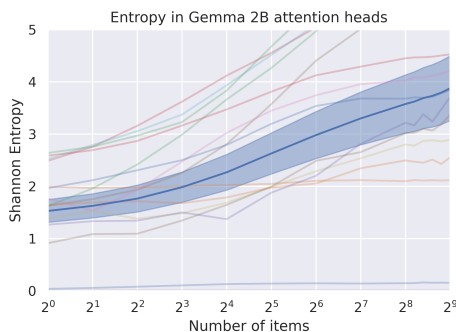

Figure 3: Entropy of attention heads in the first block of Gemma 2B with prompt `"What is the maximum in the following sequence: {seq}? The maximum is:"` and varying the number of elements in seq. Each curve is one attentional head; the blue shaded curve is the mean and standard deviation across all of them.

While the model indeed attributes focus sharply and cleanly on the maximum item, this only holds true on the problem sizes that the model was trained on. As we simulate an out-of-distribution setting where the problem size increases (without changing the value distribution), the attentional coefficients eventually disperse towards the uniform distribution.

This effects manifests in the attention heads of Transformers as well—we visualise the *entropy* (a proxy for sharpness) of Gemma 2B (Gemma Team et al., 2024)'s heads when answering a similar maximisation task in Figure 3.

In fact, we can show that this effect is *inevitable* in `softmax` using the following Lemma:

**Lemma 2.1** (`softmax` must disperse). *Let $\mathbf{e}^{(n)} \in \mathbb{R}^n$ be a collection of $n$ logits going into the* `softmax`$_\theta$ *function with temperature $\theta > 0$, bounded above and below s.t. $m \leq e_k^{(n)} \leq M$ for some $m, M \in \mathbb{R}$. Then, as more items are added ($n \to +\infty$), it must hold that, for each item $1 \leq k \leq n$,* `softmax`$_\theta(\mathbf{e}^{(n)})_k = \Theta(\frac{1}{n})$. *That is, the computed attention coefficients **disperse** for all items.*

*Proof.* Let us denote the attentional coefficient assigned to $k$ by $\alpha_k^{(n)} = $ `softmax`$_\theta(\mathbf{e}^{(n)})_k \in [0, 1]$. Then we can bound $\alpha_k^{(n)}$ above as:

$$\alpha_k^{(n)} = \frac{\exp(e_k^{(n)}/\theta)}{\sum_l \exp(e_l^{(n)}/\theta)} \leq \frac{\exp(M/\theta)}{n\exp(m/\theta)} = \frac{1}{n}\exp\left(\frac{M-m}{\theta}\right) \tag{6}$$

Similarly, we can bound $\alpha_k^{(n)}$ below as:

$$\alpha_k^{(n)} = \frac{\exp(e_k^{(n)}/\theta)}{\sum_l \exp(e_l^{(n)}/\theta)} \geq \frac{\exp(m/\theta)}{n\exp(M/\theta)} = \frac{1}{n}\exp\left(\frac{m-M}{\theta}\right) \tag{7}$$

Hence, if we let $\delta = (M - m)$

$$\frac{1}{n} \exp -\frac{\delta}{\theta} \leq \alpha_k^{(n)} \leq \frac{1}{n} \exp \frac{\delta}{\theta} \tag{8}$$

Which implies $\alpha_k^{(n)} = \Theta(\frac{1}{n})$ as $\delta$ and $\theta$ are both constants. $\qquad \square$

Lemma 2.1 relies on being able to bound the logit values with specific constants. The difference of these bounds (the *spread*, $\delta = \max_i e_i^{(n)} - \min_j e_j^{(n)}$) directly controls the rate of dispersion. In modern Transformer LLM architectures operating over a vocabulary of possible token values, we can actually bound the logits in every single attentional layer—implying that dispersion *must* happen everywhere in a Transformer for sufficient problem sizes. We prove this important result now:

**Theorem 2.2** (`softmax` in Transformers over vocabularies must disperse). *Let $\mathcal{X} \subset \mathbb{R}^m$ be a set of possible $m$-dimensional input features, and let $\mathbf{X}^{(n)} \in \mathcal{X}^n$ be a matrix of input features for $n$ items. Further, assume that input features come from a **finite** set of possible values, i.e. $|\mathcal{X}| < |\mathbb{N}|$. Let $e_j^{(n)} = (\mathbf{q}^{(n)})^\top \mathbf{k}_j^{(n)}$ where $\mathbf{q}^{(n)} = \phi(\mathbf{x}_1^{(n)}, \ldots, \mathbf{x}_n^{(n)})$ and $\mathbf{K}^{(n)} = \kappa(\mathbf{x}_1^{(n)}, \ldots, \mathbf{x}_n^{(n)})$, where $\phi : \mathcal{X}^n \to \mathbb{R}^k$ and $\kappa : \mathcal{X}^n \to \mathbb{R}^{n \times k}$ are continuous functions, each expressible as a composition of $L$ layers $g_L \circ f_L \circ \cdots \circ g_1 \circ f_1$ where each layer contains a feedforward component $f_i(\mathbf{z}_1, \ldots, \mathbf{z}_n)_k = f_i(\mathbf{z}_k)$ or a self-attentional component $g_i(\mathbf{z}_1, \ldots, \mathbf{z}_n)_k = \sum_{1 \leq l \leq n} \alpha_{lk} v_i(\mathbf{z}_l)$ where $\alpha_{lk} \in [0, 1]$ are `softmax`-normalised attention coefficients and $v_i$ is a feedforward network. Then, for any $\theta > 0$ and $\epsilon > 0$, there must exist an $n \in \mathbb{N}$ such that $\mathtt{softmax}_\theta(\mathbf{e}^{(n)})_k < \epsilon$ for all $1 \leq k \leq n$. That is, attention coefficients must **disperse** in all global Transformer heads if the input vocabulary is finite.*

*Proof.* Firstly, note that since $\mathcal{X}$ is a finite set of $m$-dimensional vectors, then it is also part of a *compact* space spanning all convex combinations of those vectors. Then, all feedforward layers, $f_i$ and $v_i$, being continuous functions, move inputs from a compact set to another compact set. Similarly, every self-attentional layer, $g_i$, computes a convex combination of the outputs of $v_i$, and as such, if outputs of $v_i$ are on a compact space, the outputs of $g_i$ remain on the same compact space. Therefore, if the input space of $\phi$ and $\kappa$ is compact, then the output space of $\phi$ and (each row of) $\kappa$ on $\mathbb{R}^k$ must be compact as well, regardless of the choice of $n$. Further, the dot product of two vectors $(\mathbf{q}^{(n)})^\top \mathbf{k}_j^{(n)}$ coming from compact spaces must be compact as well. Hence, the logits must be bounded by $m \leq e_k^{(n)} \leq M$ for constant $m$ and $M$. Then, letting $\delta = M - m$, we know (Lemma 2.1) that $\mathtt{softmax}_\theta(\mathbf{e}^{(n)})_k \leq \frac{1}{n} \exp(\delta/\theta)$, so for all $n > \frac{\exp(\delta/\theta)}{\epsilon}$ this value will be below $\epsilon$. $\qquad \square$

It might seem intuitive that attention head dispersion is a potentially destructive event, which forces the Transformer into misclassifying certain inputs. We prove this intuition in Appendix B. We also discuss the rate at which dispersion occurs at various model depths in Appendix C.

## 3 ADAPTIVE TEMPERATURE

Since we now know dispersion is inevitable, are there any ways we can leverage our theory's findings to make `softmax` sharper? One obvious constraint our theory rests on is the assumption that $\theta > 0$, i.e. that our temperature is nonzero. While zero temperature—also known as *hard attention* (Denil et al., 2012; Ranzato, 2014; Mnih et al., 2014; Xu et al., 2015)—guarantees sharpness, training large-scale Transformers with it tends to not work well in practice (Bica et al., 2024).

What about applying $\theta = 0$ to an *already-trained* Transformer? We can show this is also problematic since, for any attention head where the Transformer has learnt to induce sharpness, it *necessarily* did so by increasing magnitude of its weights (see Appendix D for a proof and numerical validation):

**Proposition 3.1** (Sharpness in Transformers necessitates large weights). *Let $\mathbf{e}^{(n)} \in \mathbb{R}^n$ be a collection of $n$ logits, computed using a dot product attention mechanism; i.e. $e_k^{(n)} = \langle \mathbf{Q}\mathbf{y}, \mathbf{K}\mathbf{x}_k \rangle$, where $\mathbf{y} \in \mathbb{R}^m$ is a query vector and $\mathbf{Q}, \mathbf{K} \in \mathbb{R}^{m' \times m}$ are parameters. Let $\delta = \max_{1 \leq i \leq n} e_i^{(n)} - \min_{1 \leq j \leq n} e_j^{(n)}$ be their maximum difference. Then $\delta$ is upper bounded as $\delta \leq 2\sigma_{\max}^{(Q)} \sigma_{\max}^{(K)} \|\mathbf{y}\| \max_{1 \leq i \leq n} \|\mathbf{x}_i\|$, where $\sigma_{\max}^{(Q)}, \sigma_{\max}^{(K)} \in \mathbb{R}$ are the largest singular values of $\mathbf{Q}$ and $\mathbf{K}$. That is, the sharpness of the softmax in Transformers depends on the norm of its parameters.*

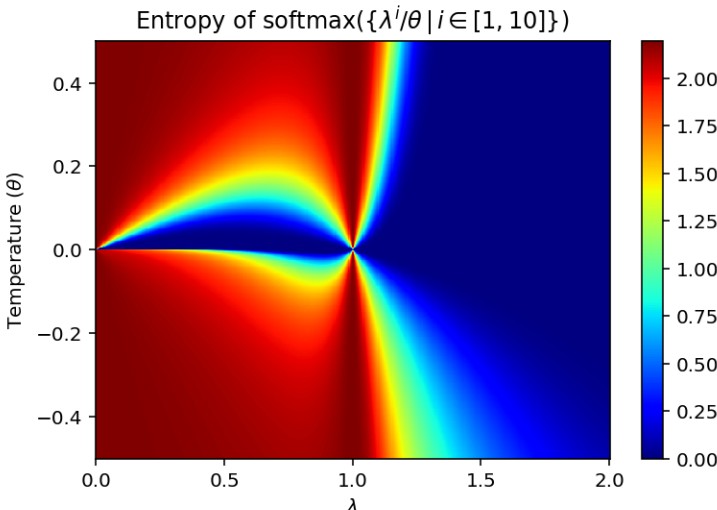

Figure 4: Entropy of the $\mathtt{softmax}_\theta$ function for 10 elements of a power series. Entropy increases with temperature but the rate at which it increases is heavily dependent on the attention logit distribution. Degenerate cases: near $\lambda = 0$ and $\lambda = 1$ all logits are the same, leading to highest entropy.

Note that there is a common practice of leveraging operators such as *layer normalisation* (Ba et al., 2016) extensively within Transformer architectures, which clamps $\|\mathbf{x}_i\|$ and $\|\mathbf{y}\|$ if applied right before the query-key mechanism, accentuating the impact of $\mathbf{Q}$ and $\mathbf{K}$'s singular values.

However, forcing large parameters promotes overfitting, and the likelihood that the *incorrect* item gets the largest logit—see Figure 2. Setting temperature to zero will then *degrade* accuracy—we might prefer to make the coefficients sharper while making sure that the chosen item is not left behind. This motivates our use of **adaptive temperature**, where we vary $\theta$ depending on the *entropy* in the input coefficients. Adaptive temperature can be elegantly motivated by the fact that decreasing the temperature must monotonically decrease the entropy, which is well-known in thermodynamics:

**Proposition 3.2** (Decreasing temperature decreases entropy). *Let $\mathbf{e}^{(n)} \in \mathbb{R}^n$ be a collection of $n$ logits. Consider the Boltzmann distribution over these $n$ items, $p_i \propto \exp(-\beta e_i^{(n)})$ for $\beta \in \mathbb{R}$, and let $H = -\sum_i p_i \log p_i$ be its Shannon entropy. Then, as $\beta$'s magnitude increases, $H$ must monotonically decrease. Thus, since $\beta \propto \frac{1}{\theta}$ where $\theta$ is the temperature in $\mathtt{softmax}_\theta$, decreasing the temperature must monotonically decrease the entropy.*

We provide a full proof in Appendix E. To supplement Proposition 3.2 empirically, we also provide—in Figure 4—a visualisation of how the Shannon entropy varies with temperature, for a 10-logit input with varying spread between the logits.

To compute the approximate temperature value as a function of entropy, we generate a dataset of inputs to our model where the maximal items do not obtain the highest logit. For each such input, we find the "optimal" value of $\theta$ that would maximise its probability. Then we fit an inverse degree-4 polynomial to this data—see Figure 5—and use it to predict temperatures to use at inference time. Note we do not wish to increase entropy; as such, we do not correct $\theta$ to values greater than 1.

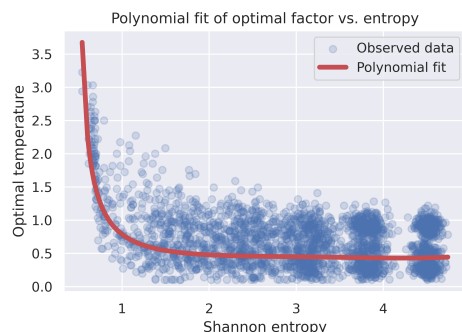

Figure 5: The polynomial fit used to derive our adaptive formula for $\theta$ as a function of the Shannon entropy, $H$. The fit degree-4 function was $\theta \approx 1/(-1.791 + 4.917H - 2.3H^2 + 0.481H^3 - 0.037H^4)$. We do not apply the correction to $\theta$ if predicted greater than 1.

Table 1: Improvements observed when applying adaptive temperature on the max retrieval task (without changing the parameters), averaged over ten seeds. $p$-values computed using a paired $t$-test.

| Model | ID size 16 | Out-of-distribution sizes | | | | | | | | | |
|---|---|---|---|---|---|---|---|---|---|---|---|
| | | 32 | 64 | 128 | 256 | 512 | $1,024$ | $2,048$ | $4,096$ | $8,192$ | $16,384$ |
| Baseline | $98.6\%$ | $97.1\%$ | $94.3\%$ | $89.7\%$ | $81.3\%$ | $70.1\%$ | $53.8\%$ | $35.7\%$ | $22.6\%$ | $15.7\%$ | $12.4\%$ |
| Adaptive $\theta$ | $98.6\%$ | $97.1\%$ | $94.5\%$ | $89.9\%$ | $82.1\%$ | $72.5\%$ | $57.7\%$ | $39.4\%$ | $24.9\%$ | $17.5\%$ | $14.0\%$ |
| $p$-value | $0.4$ | $0.4$ | $0.002$ | $2 \cdot 10^{-5}$ | $2 \cdot 10^{-4}$ | $3 \cdot 10^{-5}$ | $10^{-4}$ | $6 \cdot 10^{-4}$ | $0.02$ | $10^{-3}$ | $4 \cdot 10^{-3}$ |

The JAX (Bradbury et al., 2018) implementation of our adaptive-$\theta$ softmax is provided below, and we use it as a drop-in replacement for `jax.nn.softmax` in all of our experiments.

```python
def adaptive_temperature_softmax(logits):
  original_probs = jax.nn.softmax(logits)

  poly_fit = jnp.array([-0.037, 0.481, -2.3, 4.917, -1.791])  # see Figure 5
  entropy = jnp.sum(-original_probs * jnp.log(original_probs + 1e-9),
                    axis=-1, keepdims=True)  # compute the Shannon entropy
  beta = jnp.where(  # beta = 1 / theta
      entropy > 0.5,  # don't overcorrect low-entropy heads
      jnp.maximum(jnp.polyval(poly_fit, entropy), 1.0),  # never increase entropy
      1.0)

  return jax.nn.softmax(logits * beta)
```

While this approach requires two calls to `jax.nn.softmax` in place of one, as well as computing several additional intermediate tensors, we are able to implement it in a way that allows the entropy correction computation to be fully *streamed*, and hence compatible with efficient, scalable approaches like Flash Attention (Dao et al., 2022) that uses $O(n)$ rather than $O(n^2)$ memory to compute attention. We provide the derivation of our streamed algorithm in Appendix F.

Note we are not the first to propose dynamically adapting temperature—Neumann et al. (2018); Radford et al. (2021) do this in the classification layer (and hence do not have to handle an ever-increasing amount of items), whereas Chiang & Cholak (2022); Cao et al. (2024) perform it over intermediate attentional heads, but in a way that only depends on problem size (e.g. multiplying logits by $\log n$), hence not taking into account initial logit sharpness. It is important to also call out AERO (Jha & Reagen, 2024), a method which introduces *learnable* temperature, and Entropix (xjdr & doomslide, 2024), a notable library for (var)entropy-based LLM sampling.

## 4 EXPERIMENTAL RESULTS

To validate the utility of our proposed adaptive temperature scheme, we evaluate it on both our previously-mentioned max retrieval task—which allows us a pristine environment for evaluating whether adaptive temperature leads to more useful attention heads—as well as the CLRS-Text algorithmic reasoning benchmark (Markeeva et al., 2024), which represents a challenging reasoning task for decoder-only Transformers, and is hence likely to require low-entropy behaviour.

### 4.1 max RETRIEVAL

For this task, we first train our single attention head architecture as described in Appendix A; then, we evaluate it at various numbers of input items, with and without applying adaptive temperature to its sole softmax function call. Note that this is a "pure" inference time adjustment—no modifications to the learned parameters are performed!

The results—averaged over ten seeds and with statistical significance tests applied—are summarised in Table 1. As is evident, applying adaptive temperature leads to a more performant retrieval head on out-of-distribution inputs, with statistical significance ascertained via a paired $t$-test.

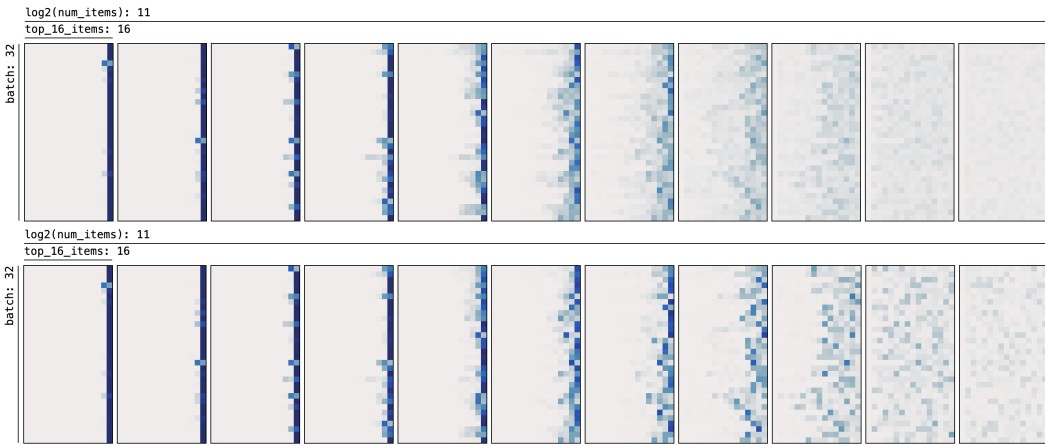

Figure 6: Visualising the attentional head for the max retrieval task with (**below**) and without (**above**) adaptive temperature applied, for the same batch and parameters as in Figure 2. Note the increased sharpness in the coefficients, especially as the amount of items increases.

These results are further supplemented by a qualitative comparison of the softmax coefficients before and after applying the temperature adaptation. As can be seen in Figure 6, our proposed adaptive temperature adaptation indeed leads to sharper coefficients out-of-distribution and higher attention being directed to the desired item, even in situations where it did not receive the largest logit.

We have now successfully validated the predictions of our theory in a controlled environment. What about a more challenging benchmark with a baseline model comprising *many* attentional heads?

## 4.2 CLRS-Text

In this benchmark, we follow the protocol established by Markeeva et al. (2024) and fine-tune Gemma 2B models (Gemma Team et al., 2024) on the thirty algorithmic execution tasks in CLRS-Text, plotting their performance profiles in- and out-of-distribution at various problem sizes.

While it may be tempting to directly re-apply our learned adaptive temperature function from Figure 5 solely at inference time—the same way we did in the max retrieval experiments—this approach does not empirically work well in the CLRS-Text regime. This is due to the fact that CLRS-Text inputs are often textual representations of *floating-point* numbers and therefore individual numbers often span *multiple* tokens. It is therefore insufficient and inappropriate to aim for entropy levels where all the focus would be on *one* token only, as was desirable in the max retrieval task.

One follow-up on this could be to perform exactly the same polynomial fit exercise leading up to Figure 5, only this time focussing on "optimal" values of temperature for Gemma's attentional heads. However, in this regime, we argue this exercise is substantially less trivial to do—as we are now dealing with a system spanning many attentional heads across many layers, it is not easy to even discover relevant attentional heads' behaviours, and even less so to ascertain that the model's robustness depends on those specific heads in those ways. As briefly discussed before, any such individual endeavour typically leads to a brand-new research project in mechanistic interpretability, and we do not find this to be in-scope of our paper.

That being said, there is an alternate route to make the Gemma model still benefit from our adaptive temperature module exactly as-is (i.e., with exactly the same polynomial fit as in Figure 5); it just has to directly *learn* how to leverage it. As such, in our CLRS-Text ablation we apply adaptive temperature both during fine-tuning and at inference time. What this means is, we replace all instances of jax.nn.softmax within all the attentional heads of Gemma 2B with our adaptive_temperature_softmax function, both during fine-tuning of the model and during inference. This allows the model to learn how to compute key/query embeddings that can maximally exploit the temperature adaptation.

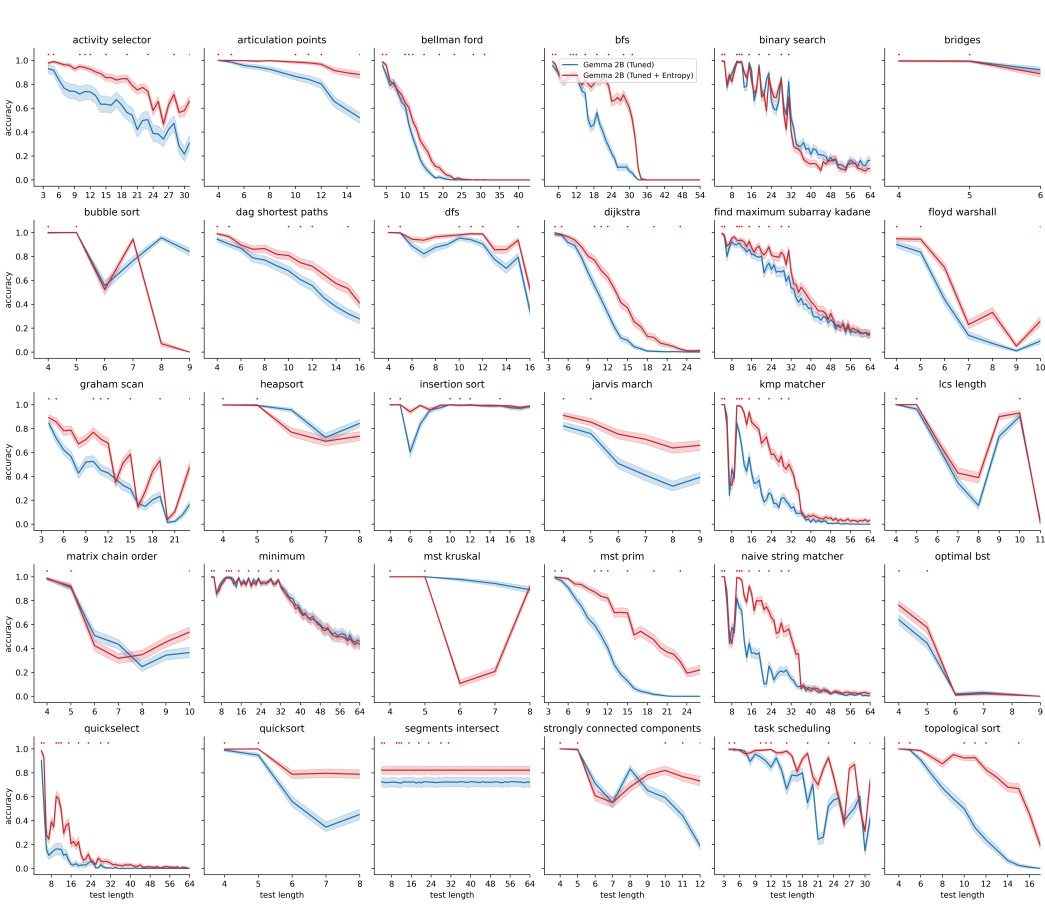

Figure 7: Resampling test results on CLRS-Text of variants of Gemma 2B, fine-tuned with and without adaptive temperature applied, on various problem sizes. Each point on the $x$ axis corresponds to a particular problem size in the corresponding algorithmic task. For example, on sorting tasks, this corresponds to the number of items being sorted; for graph tasks, it corresponds to the number of nodes in the graph. The **blue** curves represent the accuracy of the baseline fine-tuned Gemma 2B model, whereas the **red** curves represent the accuracy of that same model, fine-tuned with adaptive temperature. Both Gemma 2B variants were explicitly trained on CLRS-Text tasks—the training set sizes are denoted by red dots—and are evaluated zero-shot. Note that we limit our sample length to $2,048$ tokens, and only show performance metrics for sizes where the answer fits in this constraint.

These final comparative results may be found in Figure 7, and they demonstrate a significant advantage of the adaptive temperature-backed model on nearly all of the thirty algorithms study. This indicates that, even in a complex system with many interactions between attentional heads, it is possible to extract benefits from the simple idea of dynamically adapting the temperature—and we hope our result paves the way for more involved future investigation of such approaches.

## 5 CONCLUSIONS

*"Energy continuously flows from being concentrated*
*To becoming **dispersed**, spread out, wasted and useless."*—The 2nd Law: Unsustainable, by Muse

In this paper, we have provided extensive theoretical and empirical evidence that the softmax—a key function in the design of modern frontier architectures—is fundamentally unable to sustain robust reasoning behaviours across all possible inputs, as its output coefficients are necessarily dispersing provided sufficient input elements.

Beyond illustrating and proving these dispersion effects, we also attempted to use our theoretical framework to propose an *adaptive temperature* approach that is able—at least to a certain extent—to hold the dispersion effect at bay. It is our opinion that the favourable results we observe with adaptive temperature warrant further investigation, and indicate that such adaptive layers are a strategy worth dedicating further attention to in future work.

We conclude by remarking, once again, that adaptive temperature is merely an *ad-hoc* method and it does not escape the conclusions of our theory! The key takeaway of our paper is *not* the adaptive temperature proposal; it is the fact that we find it worthwhile to more seriously invest in research of hybrid architectures that will not fully rely on the softmax function, at least within the confines of the assumptions of our theory. To name a few possibilities:

- Any kind of unnormalised attention, such as *linear* (Schmidhuber, 1992), *sigmoidal* (Ramapuram et al., 2024) or *stick-breaking* attention (Tan et al., 2024) does not have the dispersion issues presented here. That being said, it becomes substantially harder to meaningfully *rank* items using them, see e.g. the GATv2 paper (Brody et al., 2022).

- Similarly, forcing the attention to be *hard* or *local* (Martins & Astudillo, 2016; Correia et al., 2019; Peters et al., 2019) would also escape the confines of our theory. We already briefly discussed the challenges of learning with hard attention—local attention provides a very interesting alternative, but it must be stressed that "out-of-distribution" behaviours for certain heads may appear even at highly "local" scales; OOD here refers to going outside *the largest problem size the head saw at training time*, **not** the largest context deployed at training time.

- Lastly, our key Theorem relies on the model being built out of *continuous* building blocks. Inserting *discontinuities* in the feedforward layers—perhaps using approaches like Dudzik et al. (2024) as inspiration—would also break the assumptions of our theory, though it comes with obvious challenges to learning at scale.

While such approaches haven't seen as much success at scale as the "vanilla" Transformer, we hope our results inspire future work into making them stable, especially for constructing reasoning systems.

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

## A    EXPERIMENTAL DETAILS FOR THE MAXIMUM ENTRY RETRIEVAL TASK

As briefly described in the main paper, we leverage the max retrieval task over a single attention head as a way to empirically validate our theory, as well as assess the benefits of adaptive temperature in a controlled setting. In this section, we describe the various aspects of our experimental setup, for the purposes of clarity and reproducibility.

### A.1    MOTIVATION

We deliberately focus on a *single attention head* environment and a *simple* selection function (max) to remove any confounders from our observations.

Since we are using exactly one attention head, whatever coefficients it outputs can be directly related to the network's belief in which items are most important for the downstream prediction. This allows us to, e.g., correlate the coefficients with the ground-truth magnitude of the items.

Since we are looking for the maximal element's property, we are not requiring any complicated behaviour from the coefficients: when our target task is to approximate max, the softmax coefficients need to approximate argmax—which is exactly what they are designed to be a smooth approximation for. As such, this choice of target task exhibits high algorithmic alignment (Xu et al., 2020).

### A.2    DATA GENERATION

Let $n$ be the number of items in the set that we wish to classify. For each item, $1 \le i \le n$, we need to define a *priority value*, which is used to select the maximal entry. We sample these values from a uniform distribution; $\rho_i \sim \mathcal{U}(0, 1)$.

We would also wish our task to be a *classification* rather than *regression* task, in order to leverage a more robust accuracy metric. As such, let $C$ be the desired number of classes. We can now attach to each item a class, $\kappa_i \sim \mathcal{U}\{1, \dots, C\}$, sampled uniformly at random. We assume $C = 10$ fixed.

Then, for each input item, $1 \le i \le n$, we consider its features to be $\mathbf{x}_i \in \mathbb{R}^{C+1}$ to be defined as $\mathbf{x}_i = \rho_i \| \text{onehot}(\kappa_i, C)$, i.e. the concatenation of these two sampled pieces of data where $\kappa_i$ is represented as a one-hot vector.

Lastly, since we will leverage dot-product attention, we also need a *query* vector. In this particular task, the query is irrelevant, and we initialise it to a random uniformly-sampled value, $q \sim \mathcal{U}(0, 1)$.

Our task is to predict, given $\{\mathbf{x}_i\}_{1 \le i \le n}$ and $q$, the class of the maximal item, i.e., $\kappa_{\arg\max_i \rho_i}$.

## A.3 NEURAL NETWORK ARCHITECTURE

The neural network model is designed to be a simple set aggregation model (in the style of Deep Sets (Zaheer et al., 2017)), with a single-head dot product attention as the aggregation function.

Its equations can be summarised as follows:

$$\mathbf{h}_i = \psi_x(\mathbf{x}_i) \tag{9}$$

$$\mathbf{q} = \psi_q(q) \tag{10}$$

$$e_i = (\mathbf{Q}\mathbf{q})^\top (\mathbf{K}\mathbf{h}_i) \tag{11}$$

$$\alpha_i = \frac{\exp(e_i/\theta)}{\sum_{1 \le j \le n} \exp(e_j/\theta)} \tag{12}$$

$$\mathbf{z} = \sum_{1 \le i \le n} \alpha_i \mathbf{V}\mathbf{h}_i \tag{13}$$

$$\mathbf{y} = \phi(\mathbf{z}) \tag{14}$$

Equations 2–3 prepare the embeddings of the items and query, using two-layer MLPs $\psi_x$ and $\psi_q$ using the GeLU activation function (Hendrycks & Gimpel, 2016) and an embedding size of 128 dimensions. Then, a single-head dot-product attention (with query, key and value matrices $\mathbf{Q}$, $\mathbf{K}$ and $\mathbf{V}$) is executed in equations 4–6. Lastly, the output class logits are predicted from the attended vector using a two-layer GeLU MLP, $\phi$. Each component is a two-layer MLP to ensure it has universal approximation properties.

A concise implementation of our network using JAX (Bradbury et al., 2018) and Flax (Heek et al., 2024) is as follows:

```python
import jax.numpy as jnp
from flax import linen as nn
from typing import Callable

class Model(nn.Module):
  n_classes: int = 10
  n_feats: int = 128
  activation: Callable = nn.gelu

  @nn.compact
  def __call__(self, x, q):
    x = nn.Dense(features=self.n_feats)(x)
    x = self.activation(x)
    x = nn.Dense(features=self.n_feats)(x)
    x = self.activation(x)
    q = nn.Dense(features=self.n_feats)(q)
    q = self.activation(q)
    q = nn.Dense(features=self.n_feats)(q)
    x = nn.MultiHeadDotProductAttention(
        num_heads=1,
        qkv_features=self.n_feats)(
        inputs_q=q,
        inputs_kv=x)
    x = nn.Dense(features=self.n_feats)(jnp.squeeze(x, -2))
    x = self.activation(x)
    x = nn.Dense(features=self.n_classes)(x)
    return x
```

## A.4 EXPERIMENTAL HYPERPARAMETERS

We train our model for $100,000$ gradient steps using the Adam SGD optimiser (Kingma & Ba, 2015) with initial learning rate of $\eta = 0.001$. At each step, we present to the model a batch of 128 input

sets. All sets within a batch have the same size, sampled uniformly from $n \sim \mathcal{U}\{5, \ldots, 16\}$. The model is trained using cross-entropy, along with $L_2$ regularisation with hyperparameter $\lambda = 0.001$.

The mixed-size training is a known tactic, designed to better prepare the model for distribution shifts on larger sets at inference time. Similarly, the weight decay follows the recommendation in Proposition 3.1, as an attempt to mitigate overfitting out-of-distribution as a byproduct of sharpening the `softmax` coefficients.

Both methods prove to be effective in deriving a stable baseline model.

## B   DISPERSION HARMS REASONING PERFORMANCE

While it is intuitive that complete coefficient dispersion is an undesirable event, it may not be immediately obvious that its occurrence may have any bearing on a reasoning model's predictive power.

In this Appendix, we provide several corollaries and remarks stemming from Theorem 2.2 that concretise specific ways in which reasoning failures will occur as a consequence of dispersion.

**Corollary B.1** (Dispersion induces reasoning failures). *Let $\mathbf{X}^{(n)} \in \mathcal{X}^n$ be a matrix of input features for $n$ items, where $\mathcal{X}$ is a finite set of possible values. Further, assume a strict total order $<$ on the elements of $\mathcal{X}$. Assume we are solving a reasoning task to find the rank of the highest-valued row $\mathbf{x}_i^{(n)}$ in $\mathbf{X}^{(n)}$ (according to $<$), using a classifier over a trained single-head attention architecture: $g\left(\sum_{1 \leq i \leq n} \alpha_i^{(n)} f\left(\mathbf{x}_i^{(n)}\right)\right)$, where $f$ and $g$ are continuous functions implemented as feedforward MLPs, and the coefficients $\alpha_i^{(n)}$ are computed using dot-product self-attention with `softmax` normalisation (as in Appendix A). Further, assume there are no ties in the class confidences predicted by $g$ when deciding how to classify $\mathbf{X}^{(n)}$. Then, assuming any floating- or fixed-point datatype with machine epsilon $\epsilon > 0$ is used to support the architecture's data representation, it will necessarily start to make prediction errors beyond a certain number of items $n$, due to the dispersion effect.*

*Proof.* Let $K$ be the size of the vocabulary $\mathcal{X} = \{\mathbf{v}_1, \ldots, \mathbf{v}_K\}$. The reasoning task presented here is effectively a $K$-class classification problem, predicting the maximum rank in a set of values from $\mathcal{X}$. Any prediction of the architecture must be of the form $g\left(\sum_{1 \leq j \leq K} \beta_j f\left(\mathbf{v}_j\right)\right)$, with the constraints that $\beta_j \geq 0$, $\sum_{1 \leq j \leq K} \beta_j = 1$ and $\beta_j = 0$ if $\mathbf{v}_j \notin \mathbf{X}^{(n)}$.

Now, consider two specific points $\mathbf{v}_a$ and $\mathbf{v}_b$ such that $\mathbf{v}_a > \mathbf{v}_b$. The architecture, if trained properly, must classify $g(f(\mathbf{v}_a))$ into the $a$ class, and $g(f(\mathbf{v}_b))$ into the $b$ class.

Let $\mathbf{X}^{(n)}$ be an input matrix formed such that $\mathbf{x}_1^{(n)} = \mathbf{v}_a$ and $\mathbf{x}_i^{(n)} = \mathbf{v}_b$ for all $1 < i \leq n$. For such an input, the desired output class is $a$, and the prediction must be of the form $g\left(\alpha_1^{(n)} f(\mathbf{v}_a) + \left(1 - \alpha_1^{(n)}\right) f(\mathbf{v}_b)\right)$.

Since the input features come from a fixed vocabulary and are processed only using feedforward networks and self-attention layers, we can leverage the argument in Theorem 2.2 to conclude that there will be a fixed *spread* in the trained architecture, $\delta$, and further that $\alpha_i \leq \frac{1}{n} \exp \frac{\delta}{\theta}$ for all $i$.

Using this we can see that, when $n > \frac{1}{\epsilon} \exp \frac{\delta}{\theta}$, it must hold that $\alpha_1^{(n)} < \epsilon$. At this point, the value of $\alpha_1^{(n)}$ will be indistinguishable from zero, and the weighted sum will reduce to $g(f(\mathbf{v}_b))$, due to the assumed continuity of $g$ around $f(\mathbf{v}_b)$.

Hence, by previous assumptions, and by the assumption that there are no ties in the class logits in $g(f(\mathbf{v}_b))^2$, at least one of the following must be true once dispersion occurs:

- The input $\{\mathbf{v}_a, \mathbf{v}_b, \mathbf{v}_b, \ldots, \mathbf{v}_b\}$ of sufficiently large size will be misclassified into class $b$;

---

[2]This assumption is important in the case that $g(f(\mathbf{v}_b))$ gives equal logits to classes $a$ and $b$. As this is a boundary condition for the classifier, if it occurred exactly on $f(\mathbf{v}_b)$, we would not be able to guarantee that any two sets mapped to $f(\mathbf{v}_b)$ will be classified identically without sacrificing local continuity around $f(\mathbf{v}_b)$. Note that, due to floating-point rounding errors, this assumption is *rarely* broken in modern deep classifiers.

- The input $\{\mathbf{v}_b, \ldots, \mathbf{v}_b\}$ (for any size) will be misclassified.

In either case, the architecture had to have made an error. $\qquad\square$

While Corollary B.1 concerns single attention heads, note that we can leverage the setting of Theorem 2.2 to prove that such failures will occur in deep Transformers as well. We sketch this intuition:

**Remark B.2.** *Given the same task as in Corollary B.1, using a deep Transformer architecture as described in Theorem 2.2, dispersion in its attentional layers is sufficient to cause misclassifications to occur. To see why, first, assume that the models have no residual connections. The arguments for why such architectures must misclassify are subtly different depending on the Transformer model:*

- *For BERT-style Transformers, since all attention heads are global, after one dispersed layer, any sufficiently large set $\{\mathbf{v}_a, \mathbf{v}_b, \ldots, \mathbf{v}_b\}$ will have identical embeddings to a set $\{\mathbf{v}_b, \ldots, \mathbf{v}_b\}$ of the same size. After this, it is impossible to classify them differently.*

- *For GPT-style Transformers, to simplify the argument, we assume the $\mathbf{v}_a$ element is at the end of the input: $\{\mathbf{v}_b, \ldots, \mathbf{v}_b, \mathbf{v}_a\}$. In this setting, only the final token's attention head will receive the features from $\mathbf{v}_a$. If it disperses, this set will once again be indistinguishable from a set $\{\mathbf{v}_b, \ldots, \mathbf{v}_b\}$ of the same size. This argument is inspired by Barbero et al. (2024).*

*Residual connections (He et al., 2016) allow for preserving the information contained in $\mathbf{v}_a$ even across dispersed layers. However, as we have assumed all heads attending over $\mathbf{v}_a$ have dispersed, no subsequent layer will be able to meaningfully integrate this information across the set, and eventually the computation will hit the final layers' attentional heads, where the final embeddings will once again be indistinguishable across these two different sets.*

We note that the only condition on the coefficients necessary for this breakdown to occur is that they decay towards zero—the failure on sets of the kind $\{\mathbf{v}_a, \mathbf{v}_b, \mathbf{v}_b, \ldots, \mathbf{v}_b\}$ is *not* prevented even if $\alpha_1^{(n)}$ decays substantially more slowly than the other coefficients!

**Remark B.3.** *If we assume a dispersion setting where*

$$\alpha_i^{(n)} = \begin{cases} \Theta\left(\frac{\log n}{n}\right) & i = 1 \\ \Theta\left(\frac{1}{n}\right) & 1 < i \leq n \end{cases}$$

*The failure described by Corollary B.1 still applies, following exactly the same proof, i.e. eventually $\alpha_1^{(n)} < \epsilon$ for any machine epsilon value $\epsilon > 0$. Note that, as per Theorem 2.2, this situation is impossible in vocabulary-based Transformer architectures.*

## C  HOW DOES DISPERSION INTERACT WITH DEPTH?

While Theorem 2.2 concludes that dispersion must eventually affect all global attention heads in Transformer architectures over vocabularies, not much is said about how rapidly the dispersion must affect heads at various depths.

Intuitively, if dispersion occurs at a particular layer, it will cause the outputs of the dispersed attention heads to converge to the average of all value vectors. This convergence, in turn, minimises the *spread* of logits, $\delta$, that the subsequent layer will experience. As shown by Lemma 2.1, the value of the spread directly controls at which sizes dispersion will occur.

Using this argument, we can show that in BERT-style Transformers without residual connections, a complete dispersion of all heads in a particular layer leads *all* subsequent layers' attention heads to immediately disperse.

**Remark C.1.** *Let $\mathbf{H}^{(n)} = \{\mathbf{h}_i^{(n)}\}_{1 \leq i \leq n}$ be the input node embeddings for an intermediate layer of a BERT-style Transformer without residual connections. If all of this layer's attention heads have dispersed on that input, i.e. $\alpha_{ij}^{(n)} < \epsilon$ where $\epsilon$ is the machine epsilon, then* all *of that layer's output node embeddings will be equal to the average embedding, $\tilde{\mathbf{h}}_i^{(n)} = \frac{1}{n}\sum_{1 \leq j \leq n} \mathbf{V}\mathbf{h}_j^{(n)}$. Since these*

*constitute the inputs for the next layer's attention heads, we can conclude that all of the next layer's key and query vectors will be identical, namely (for any feedforward layer $f$):*

$$\tilde{\mathbf{k}}_i^{(n)} = \mathbf{K}' f\left(\frac{1}{n}\sum_{1\leq j\leq n}\mathbf{V}\mathbf{h}_j^{(n)}\right) \qquad \tilde{\mathbf{q}}_i^{(n)} = \mathbf{Q}' f\left(\frac{1}{n}\sum_{1\leq j\leq n}\mathbf{V}\mathbf{h}_j^{(n)}\right)$$

*As such, all logits of such a layer will themselves be equal to*

$$\tilde{e}_{ij} = \left(\mathbf{Q}' f\left(\frac{1}{n}\sum_{1\leq j\leq n}\mathbf{V}\mathbf{h}_j^{(n)}\right)\right)^\top \left(\mathbf{K}' f\left(\frac{1}{n}\sum_{1\leq j\leq n}\mathbf{V}\mathbf{h}_j^{(n)}\right)\right)$$

*and hence, the spread will converge to $\tilde{\delta} = 0$. Given Lemma 2.1, such a layer can only compute averages for any input size $n$, which is equivalent behaviour to full dispersion. That is, dispersion in a layer implies that **all** subsequent layers will output embeddings equivalent to fully dispersed ones.*

Note that, if we introduce residual connections in BERT-style Transformers, or leverage GPT-style Transformers, these kinds of conclusions are no longer applicable. This is because residual connections, as well as the more localised attention heads in GPT-style models, ensure that not all token embeddings will converge to the average embedding (even under dispersion). And whenever the output token embeddings of an attentional layer are not fully converged, any intermediate transformations (such as the $\mathbf{K}$ and $\mathbf{Q}$ matrices) can re-amplify $\delta$ to less dispersed levels (see also Proposition 3.1).

Note this does not mean that any global attentional layer of Transformers over finite token vocabularies will escape dispersion—Theorem 2.2 proves it is inevitable—it only means that we cannot tie the exact moment a particular layer's heads will disperse to a preceding layer's dispersion event. But the dispersion of a layer will certainly play a direct part in reducing the $\delta$ value of subsequent layers, and this may well accelerate dispersion in subsequent layers.

## D    PROOF OF PROPOSITION 3.1, WITH NUMERICAL VALIDATION

**Proposition 3.1** (Sharpness in Transformers necessitates large weights). *Let $\mathbf{e}^{(n)} \in \mathbb{R}^n$ be a collection of $n$ logits, computed using a dot product attention mechanism; i.e. $e_k^{(n)} = \langle \mathbf{Q}\mathbf{y}, \mathbf{K}\mathbf{x}_k\rangle$, where $\mathbf{y} \in \mathbb{R}^m$ is a query vector and $\mathbf{Q}, \mathbf{K} \in \mathbb{R}^{m'\times m}$ are parameters. Let $\delta = \max_{1\leq i\leq n} e_i^{(n)} - \min_{1\leq j\leq n} e_j^{(n)}$ be their maximum difference. Then $\delta$ is upper bounded as:*

$$\delta \leq 2\sigma_{\max}^{(Q)}\sigma_{\max}^{(K)}\|\mathbf{y}\| \max_{1\leq i\leq n}\|\mathbf{x}_i\|$$

*where $\sigma_{\max}^{(Q)}, \sigma_{\max}^{(K)} \in \mathbb{R}$ are the largest singular values of $\mathbf{Q}$ and $\mathbf{K}$. That is, the sharpness of the softmax in Transformers depends on the norm of its parameters.*

*Proof.* We start by showing that the largest singular values of $\mathbf{Q}$ and $\mathbf{K}$ determine the maximum stretch due to that matrix acting on $\mathbf{x} \in \mathbb{R}^m$. More precisely, we wish to show:

$$\|\mathbf{Q}\mathbf{x}\| \leq \sigma_{\max}^{(Q)}\|\mathbf{x}\| \qquad \|\mathbf{K}\mathbf{x}\| \leq \sigma_{\max}^{(K)}\|\mathbf{x}\|$$

where $\|\cdot\|$ is the Euclidean norm. Since both inequalities have the same form, we focus on $\mathbf{Q}$ w.l.o.g. Many of these statements can be derived from linear algebra textbooks (Axler, 2015). However, the proofs are short enough that we re-derive them here for clarity.

Consider the singular value decomposition (SVD) $\mathbf{Q} = \mathbf{U}\boldsymbol{\Sigma}\mathbf{V}^\top$, where $\boldsymbol{\Sigma}$ is a rectangular diagonal matrix of singular values $\sigma_i^{(Q)} \in \mathbb{R}$. As $\mathbf{U}$ and $\mathbf{V}$ are orthogonal, $\|\mathbf{U}\mathbf{x}\| = \|\mathbf{V}\mathbf{x}\| = \|\mathbf{x}\|$. Therefore, $\|\mathbf{Q}\mathbf{x}\| = \|\mathbf{U}\boldsymbol{\Sigma}\mathbf{V}^\top\mathbf{x}\| = \|\boldsymbol{\Sigma}\mathbf{v}\|$, where $\mathbf{v} = \mathbf{V}^\top\mathbf{x}$, meaning that $\|\mathbf{v}\| = \|\mathbf{x}\|$. Then we derive:

$$\|\boldsymbol{\Sigma}\mathbf{v}\| = \|\mathbf{Q}\mathbf{x}\| = \sqrt{\sum_i \left(\sigma_i^{(Q)} v_i\right)^2} \leq \sigma_{\max}^{(Q)}\sqrt{\sum_i v_i^2} = \sigma_{\max}^{(Q)}\|\mathbf{x}\|$$

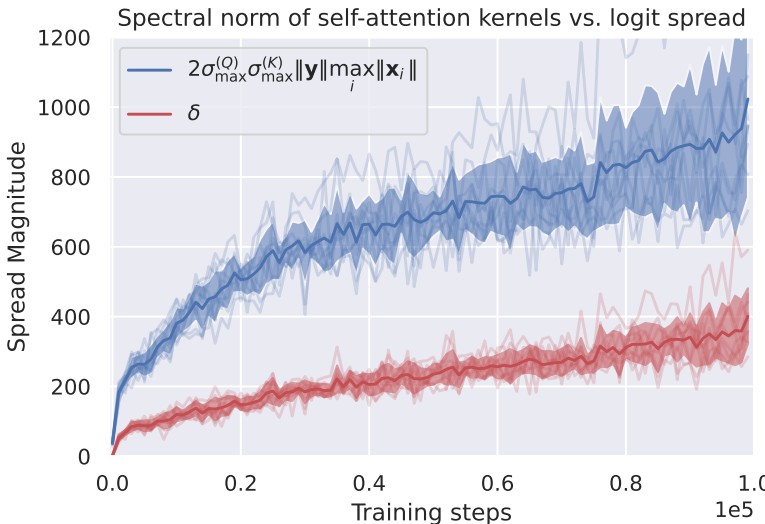

Figure 8: A plot of the logit spread, $\delta$, against its upper bound value predicted by Proposition 3.1, $2\sigma_{\max}^{(Q)}\sigma_{\max}^{(K)}\|\mathbf{y}\|\max_i\|\mathbf{x}_i\|$, for the single-head attentional experiment described in Appendix A, with statistics computed across ten seeds. This numerically validates Proposition 3.1.

We now note that

$$e_k^{(n)} = \langle \mathbf{Q}\mathbf{y}, \mathbf{K}\mathbf{x}_k \rangle = \|\mathbf{Q}\mathbf{y}\|\|\mathbf{K}\mathbf{x}_k\|\cos\theta$$

with $\theta$ the angle between the arguments of the inner product. We can now bound $e_k^{(n)}$ from above:

$$e_k^{(n)} \leq \|\mathbf{Q}\mathbf{y}\|\|\mathbf{K}\mathbf{x}_k\| \leq \sigma_{\max}^{(Q)}\sigma_{\max}^{(K)}\|\mathbf{y}\|\|\mathbf{x}_k\|$$

with $\sigma_{\max}^{(Q)}, \sigma_{\max}^{(K)}$ being the maximum singular value of $\mathbf{Q}$ and $\mathbf{K}$, respectively, and where the last step comes from the inequality shown above. Similarly, we obtain a lower bound, yielding:

$$-\sigma_{\max}^{(Q)}\sigma_{\max}^{(K)}\|\mathbf{y}\|\|\mathbf{x}_k\| \leq e_k^{(n)} \leq \sigma_{\max}^{(Q)}\sigma_{\max}^{(K)}\|\mathbf{y}\|\|\mathbf{x}_k\|$$

This gives us the desired upper bound for $\delta$:

$$\delta = \max_{1\leq i\leq n} e_i^{(n)} - \min_{1\leq j\leq n} e_j^{(n)}$$

$$\leq \max_{1\leq i\leq n} \sigma_{\max}^{(Q)}\sigma_{\max}^{(K)}\|\mathbf{y}\|\|\mathbf{x}_i\| - \min_{1\leq j\leq n} -\sigma_{\max}^{(Q)}\sigma_{\max}^{(K)}\|\mathbf{y}\|\|\mathbf{x}_j\|$$

$$= \sigma_{\max}^{(Q)}\sigma_{\max}^{(K)}\|\mathbf{y}\|\max_{1\leq i\leq n}\|\mathbf{x}_i\| + \sigma_{\max}^{(Q)}\sigma_{\max}^{(K)}\|\mathbf{y}\|\max_{1\leq j\leq n}\|\mathbf{x}_j\|$$

$$= 2\sigma_{\max}^{(Q)}\sigma_{\max}^{(K)}\|\mathbf{y}\|\max_{1\leq i\leq n}\|\mathbf{x}_i\|$$

$$\square$$

We remark that Proposition 3.1 lends itself to simple numerical verification as well. Accordingly, in Figure 8, we visualise the evolution of the logit spread, as well as its predicted upper bound, as our single-head attentional model from Appendix A is trained for increasing numbers of steps.

Indeed, we find that the upper bound is valid, and reveal a key mechanism in which our single-head architecture gradually learns to sharpen its attention: the logit spread grows with training time, but so does the norm of the relevant vectors and parameter matrices (in spite of our weight decay loss).

## E  PROOF OF PROPOSITION 3.2

**Proposition 3.2** (Decreasing temperature decreases entropy). *Let $\mathbf{e}^{(n)} \in \mathbb{R}^n$ be a collection of $n$ logits. Consider the Boltzmann distribution over these $n$ items, $p_i \propto \exp(-\beta e_i^{(n)})$ for $\beta \in \mathbb{R}$,*

*and let $H = -\sum_i p_i \log p_i$ be its Shannon entropy. Then, as $\beta$'s magnitude increases, $H$ must monotonically decrease. Thus, since $\beta \propto \frac{1}{\theta}$ where $\theta$ is the temperature in $\mathtt{softmax}_\theta$, decreasing the temperature must monotonically decrease the entropy.*

*Proof.* We start by briefly acknowledging the extremal values of $\beta$: at $\beta = 0$ (i.e., $\theta \to \infty$), all logits are weighed equally, hence $p_i = \mathcal{U}(n)$ are uniform, and entropy is maximised. Similarly, at $\beta \to \pm\infty$ (i.e., $\theta = 0$), either the minimum or the maximum logit is given a probability of 1, leading to a distribution with minimal (zero) entropy.

Now, consider the partition function $Z = \sum_i \exp(-\beta e_i^{(n)})$, such that $p_i = \frac{\exp(-\beta e_i^{(n)})}{Z}$. We will take derivatives of $\log Z$ with respect to $\beta$. Starting with the first derivative:

$$\frac{d}{d\beta} \log Z = \frac{1}{Z} \sum_i -e_i^{(n)} \exp(-\beta e_i^{(n)}) = -\sum_i e_i^{(n)} p_i = -\mathbb{E}_{i \sim p_i}(e_i^{(n)})$$

we recover the expected logit value sampled under the distribution. Now we differentiate again:

$$\frac{d^2}{d\beta^2} \log Z = -\frac{d}{d\beta} \sum_i e_i^{(n)} p_i$$

$$= -\sum_i e_i^{(n)} \frac{d}{d\beta} \frac{\exp(-\beta e_i^{(n)})}{Z}$$

$$= -\sum_i e_i^{(n)} \frac{-e_i^{(n)} \exp(-\beta e_i^{(n)}) Z - \exp(-\beta e_i^{(n)}) \sum_j -e_j^{(n)} \exp(-\beta e_j^{(n)})}{Z^2}$$

$$= \sum_i (e_i^{(n)})^2 \frac{\exp(-\beta e_i^{(n)})}{Z} - \sum_j e_j^{(n)} \frac{\exp(-\beta e_j^{(n)})}{Z} \frac{\sum_k e_k^{(n)} \exp(-\beta e_k^{(n)})}{Z}$$

$$= \sum_i (e_i^{(n)})^2 p_i - \sum_j e_j^{(n)} p_j \sum_k e_k^{(n)} p_k$$

$$= \mathbb{E}_{i \sim p_i}((e_i^{(n)})^2) - \mathbb{E}_{i \sim p_i}(e_i^{(n)})^2 = \mathrm{Var}_{i \sim p_i}(e_i^{(n)})$$

and we recover the variance of the expected logit value.

Now we turn our attention to the entropy formula:

$$H = -\sum_i p_i \log p_i = -\sum_i p_i (\log \exp(-\beta e_i^{(n)}) - \log Z)$$

$$= \sum_i p_i \log Z - \sum_j -\beta e_j^{(n)} p_j$$

$$= \log Z + \beta \mathbb{E}_{i \sim p_i}(e_i^{(n)}) = \log Z - \beta \frac{d}{d\beta} \log Z$$

To check the monotonicity of $H$ as $\beta$ varies, we now take the derivative of this expression w.r.t. $\beta$:

$$\frac{dH}{d\beta} = \frac{d}{d\beta} \log Z - \frac{d}{d\beta} \log Z - \beta \frac{d^2}{d\beta^2} \log Z = -\beta \frac{d^2}{d\beta^2} \log Z = -\beta \mathrm{Var}_{i \sim p_i}(e_i^{(n)})$$

Since variance can never be negative, we find that $\frac{dH}{d\beta} \le 0$ when $\beta \ge 0$, and $-\frac{dH}{d\beta} \le 0$ when $\beta \le 0$. As such, as the magnitude $|\beta|$ grows, the value of $H$ must monotonically decrease. $\qquad \square$

## F  AN ALGORITHM FOR STREAMING ATTENTIONAL ENTROPY

Computing our proposed adaptive temperature requires computing the entropy of the attentional coefficients. A naïve algorithm for doing so requires fully materialising the $\alpha_{ij}$ entries of the attention coefficient matrix, which requires $O(n^2)$ memory and poses scalability concerns. Fortunately, there exists an *online* algorithm for computing the entropy that is not FLOP/s efficient but does not

leverage any additional memory, allowing for a linear-space attention implementation in conjunction with Flash Attention (Dao et al., 2022). We present one such algorithm in this section. We have successfully implemented this algorithm and numerically verified that its outputs match the expected adaptive temperature amounts, allowing us to deploy layers with large context windows (up to $131,072$ tokens) on a single NVIDIA A100 node.

In order to compute the adaptive temperature, we need to first compute the attentional coefficient entropy for each row of the attentional matrix. For convenience, let us define the exponentiated logit of token $i$'s attention over token $j$, taking into account only the first $1 \le N \le n$ items:

$$\lambda_{ij}^{(N)} = \exp\left(\mathbf{q}_i^\top \mathbf{k}_j - \max_{k<N}(\mathbf{q}_i^\top \mathbf{k}_k)\right)$$

where $\mathbf{q}_i$ and $\mathbf{k}_i$ the query and key vectors, respectively, for token $i$.

Now, we can rearrange the terms of the expression for the *entropy*, $H_i^{(N)}$, of each row of the corresponding matrix of attentional coefficients, taking into account the first $N$ items, in a form that will be more favourable for streaming:

$$H_i^{(N)} = H\left(\left\{\frac{\lambda_{ij}^{(N)}}{\sum_k \lambda_{ik}^{(N)}}\right\}_{1 \le j \le n}\right)$$

$$= \sum_j \frac{\lambda_{ij}^{(N)}}{\sum_k \lambda_{ik}^{(N)}} \log \frac{\lambda_{ij}^{(N)}}{\sum_k \lambda_{ik}^{(N)}}$$

$$= \sum_j \frac{\lambda_{ij}^{(N)}}{\sum_k \lambda_{ik}^{(N)}} \left(\log \lambda_{ij}^{(N)} - \log\left(\sum_k \lambda_{ik}^{(N)}\right)\right)$$

$$= \sum_j \frac{\lambda_{ij}^{(N)}}{\sum_k \lambda_{ik}^{(N)}} \log \lambda_{ij}^{(N)} - \sum_j \frac{\lambda_{ij}^{(N)}}{\sum_k \lambda_{ik}^{(N)}} \log\left(\sum_k \lambda_{ik}^{(N)}\right)$$

$$= \frac{\sum_j \lambda_{ij}^{(N)} \log \lambda_{ij}^{(N)}}{\sum_k \lambda_{ik}^{(N)}} - \frac{\sum_j \lambda_{ij}^{(N)}}{\sum_k \lambda_{ik}^{(N)}} \log\left(\sum_k \lambda_{ik}^{(N)}\right)$$

$$= \frac{\sum_j \lambda_{ij}^{(N)} \log \lambda_{ij}^{(N)}}{\sum_k \lambda_{ik}^{(N)}} - \log\left(\sum_k \lambda_{ik}^{(N)}\right)$$

Next, we define two cumulative quantities:

$$\Lambda_i^{(N)} := \sum_{j<N} \lambda_{ij}^{(N)} \qquad m_i^{(N)} := \max_{j<N} \mathbf{q}_i^\top \mathbf{k}_j$$

which allow us to further analyse the $\sum_j \lambda_{ij}^{(N)} \log \lambda_{ij}^{(N)}$ term as follows:

$$\sum_{j<N} \lambda_{ij}^{(N)} \log \lambda_{ij}^{(N)} = \sum_{j<N} \exp\left(\mathbf{q}_i^\top \mathbf{k}_j - \max_k \mathbf{q}_i^\top \mathbf{k}_k\right) \log \exp\left(\mathbf{q}_i^\top \mathbf{k}_j - \max_k \mathbf{q}_i^\top \mathbf{k}_k\right)$$

$$= \sum_{j<N} \lambda_{ij}^{(N)} \left(\mathbf{q}_i^\top \mathbf{k}_j - m_i^{(N)}\right)$$

$$= \sum_{j<N} \lambda_{ij}^{(N)} \mathbf{q}_i^\top \mathbf{k}_j - m_i^{(N)} \Lambda_i^{(N)}$$

Now we remark that we can incrementally compute $\Lambda_i^{(N)}$ using the following iterative formula, leveraging the same concepts as Flash Attention (Dao et al., 2022):

$$\Lambda_i^{(N)} := \sum_{j<N} \lambda_{ij}^{(N)} = \sum_{j<N} \exp\left(\mathbf{q}_i^\top \mathbf{k}_j - m_i^{(N)}\right)$$

$$\Lambda_i^{(N+1)} = \Lambda_i^{(N)} \exp\left(m_i^{(N)} - m_i^{(N+1)}\right) + \lambda_{iN}^{(N)}$$

and we can incrementally compute the remaining term, $\mathcal{K}_i^{(N)} = \sum_{j<N} \lambda_{ij}^{(N)} \mathbf{q}_i^\top \mathbf{k}_j$, using the following iterative formula:

$$\mathcal{K}_i^{(N)} := \sum_{j<N} \lambda_{ij}^{(N)} \mathbf{q}_i^\top \mathbf{k}_j = \sum_{j<N} \exp\left(\mathbf{q}_i^\top \mathbf{k}_j - m_i^{(N)}\right) \mathbf{q}_i^\top \mathbf{k}_j$$

$$\mathcal{K}_i^{(N+1)} = \mathcal{K}_i^{(N)} \exp\left(m_i^{(N)} - m_i^{(N+1)}\right) + \lambda_{iN}^{(N)} \mathbf{q}_i^\top \mathbf{k}_N$$

So our final result in terms of $\Lambda_i^{(n)}$ and $\mathcal{K}_i^{(n)}$ (fully streamed across all $n$ items) is:

$$H_i^{(n)} = \frac{\mathcal{K}_i^{(n)} - m_i^{(n)} \Lambda_i^{(n)}}{\Lambda_i^{(n)}} - \log \Lambda_i^{(n)}$$

$$= \frac{\mathcal{K}_i^{(n)}}{\Lambda_i^{(n)}} - m_i^{(n)} - \log \Lambda_i^{(n)}$$

This expression can be computed with $O(n)$ memory, as we never have to materialise an entire matrix of coefficients. Under this implementation, adaptive temperature can easily scale to large context windows (which we have validated empirically up to $131,072$ tokens).

