# OpenReview forum: "softmax is not enough (for sharp out-of-distribution)"
_ICLR.cc/2025/Conference — Submitted to ICLR 2025_

### Official Review · Reviewer_4NvJ · 2024-10-23

**Soundness:** 3
**Presentation:** 1
**Contribution:** 2
**Rating:** 6
**Confidence:** 4

**Summary:**

The authors show that at inference time, the post-softmax of the self-attention layer will disperse as the input sequence length grows to infinity. This is due to the fact that softmax cannot approximate "sharpness" when the input is bounded. To address this limitation, the authors propose an inference-time procedure termed adaptive temperature, and conduct experiments on max retrieval and CLRS Text to validate its effectiveness.

**Strengths:**

- The topic of study is interesting
- The discussion in Section 5 could potentially inspire future research

**Weaknesses:**

There are at least two main weaknesses:

- The presentation is unsatisfactory in several ways: 1) There is no preliminary section introducing the basics of transformers (and others), and all notations are squeezed into the statement of Theorem 1; 2) Many unconventional terms, such as "attentional head" and "attention coefficients", are used frequently without definitions or explanations; 3) The figure captions are overly brief and impede understanding, for example, it is unclear what "batch" refers to or what each row represents in Figure 2, the meaning of the different curves and the shaded blue region in Figure 3 is not explained, and the x-axis in Figure 7 is unclear, with the legend appearing only in one small figure (which could easily be overlooked); 4) The procedure for applying adaptive temperature is not formally described, which is necessary given that multiple self-attention modules are present across different layers in language models.

- The paper covers theory, algorithms, and experiments, but none of these components seem to be particularly strong, making it difficult to identify the main contribution of the paper.
  - **Theory**. The main result, Theorem 2, seems to be a straightforward corollary of Lemma 1 (relying on the fact that the continuous mapping of a compact set remains compact), which itself leverages a basic property of softmax. While I’m not advocating for fancy proof techniques, the real concern is that the conclusion of Theorem 2 feels unsatisfying. Specifically, 1) it is unclear what the consequence of such "dispersion" of attention coefficients is: does it imply the failure of the underlying reasoning task? Will it still be problematic if the ground truth token has a coefficient of $O(\frac{\log n}{n})$ while the other tokens have coefficients of $O(\frac{1}{n})$, where their *ratio* still goes to infinity? 2) The statement is too broad and applies equally to any self-attention module in a language model; a more interesting question might be whether self-attention modules in deeper (i.e., later) layers suffer more from this dispersion phenomenon.
  - **Algorithm**. The authors themselves acknowledge that adaptive temperature does not fundamentally address the dispersion issue, which is reflected in the experimental results. For example, in Table 1, the improvement over the simple baseline is not very significant.
  - **Experiments**. The paper mainly focuses on the max retrieval task. For the CLRS-Text benchmark, the authors adjusted their algorithm by applying adaptive temperature both at inference time and during fine-tuning. However, it’s unclear where the performance gains come from. Is approximating sharpness still relevant for these tasks? More broadly, what is the implication of the paper's results for general reasoning tasks?

While I appreciate the idea and topic of the study, the paper needs to address the presentation issues and strengthen **one** of the three aspects to be considered for acceptance. Note I think it is perfectly fine that a paper does not have significant algorithmic contributions.

**Minor**: Although I’m not super familiar with the related work, two lines of research, length generalization and attention sink, could be relevant.

------------
**Post-rebuttal**: Acknowledging the authors' response and the revisions to the paper, I am increasing my score to 6. That said, I feel it is important to note that the changes made are substantial and should be taken into account when AC/SAC make their final decision.

**Questions:**

See above.

---

> ### Author Response · Authors · 2024-11-21
> **Reply to Reviewer 4NvJ (Part I)**
>
> Dear Reviewer 4NvJ,
>
> For starters, we would like to give you particular thanks – while your review was negative in its rating, we’ve been able to clearly detect positive sentiment about our work, and highly pointed suggestions for how to improve our work. Many of these suggestions were indeed very much called for.
>
> We truly believe that, as a result, we now have a much stronger contribution on our hands – and we hope you will agree! The paper has already been revised on OpenReview. We summarise specific things we have done in response:
>
> ### **Improving the presentation**
>
> We believe that we have now addressed all of your remarks on presentation:
>
> * A full primer on Transformers (which also defines attentional heads and coefficients) has been added within our Introduction section. We also more precisely defined dispersion, sharpness and the specific out-of-distribution generalisation setting we study in response to the other reviewers – all of these are in the Introduction.
> * The captions of Figures 2, 3, and 7 have all been amended to properly define the various terms you called out: batch / rows (Fig. 2), curves and shaded blue region (Fig. 3), x-axis and individual curves (Fig. 7).
>
> In addition, we now have a more proper description of how adaptive temperature is applied, both in the main body of text:
>
> > That being said, there is an alternate route to make the Gemma model still benefit from our adaptive temperature module exactly as-is (i.e., with exactly the same polynomial fit as in Figure 5); it just has to directly \emph{learn} how to leverage it. As such, in our CLRS-Text ablation we apply adaptive temperature both during fine-tuning and at inference time. What this means is, we replace all instances of \texttt{jax.nn.softmax} within all the attentional heads of Gemma 2B with our \texttt{adaptive\_temperature\_softmax} function, both during fine-tuning of the model and during inference. This allows the model to learn how to compute key/query embeddings that can maximally exploit the temperature adaptation.
>
> Further, in Appendix F, where we now detailedly describe a – to the best of our knowledge – novel algorithm for computing the entropy values in a streaming manner (akin to Flash Attention), allowing us to compute adaptive temperature in $O(n)$ space complexity and deploy it on large context lengths – up to 131,072 tokens on a single A100 node.
>
> ### **Improving the theoretical conclusions**
>
> As already mentioned, we really appreciate your pointed feedback. As such, we will dedicate the majority of our rebuttal response to you on how we improved the theoretical conclusions in the previous couple of weeks—we believe we have managed to meaningfully address all of your questions. _(For what it’s worth, it is also our belief that we’ve done useful things (e.g. the aforementioned streaming algorithm for scaling adaptive temperature) to strengthen the algorithm/experiment points as well.)_
>
> > it is unclear what the consequence of such "dispersion" of attention coefficients is: does it imply the failure of the underlying reasoning task?
>
> Yes it does! We prove (Corollary B.1 and Remark B.2 in Appendix B) that attentional dispersion will be a provable culprit of misclassifications for even rudimentary reasoning tasks such as finding maximal tokens. We elaborate a lot more in our proof, but the core idea is to leverage pairs of input sets of the form {A B B … B} (max is A) and {B … B} (max is B). Under dispersed softmax, for sufficiently many Bs, at least one of these sets must be misclassified, as soon as the dispersion decay surpasses machine epsilon. _Note that modern LLMs are often heavily quantised when deployed for fast inference, so effective machine epsilon may be significantly higher than what one might expect e.g. for single-precision floating point types._
>
> > Will it still be problematic if the ground truth token has a coefficient of $O\left(\frac{\log n}{n}\right)$ while the other tokens have coefficients of $O\left(\frac{1}{n}\right)$, where their ratio still goes to infinity?
>
> Yes (and we elaborate on this in Remark B.3). There is nothing in the proof of Corollary B.1 that depends on the rate of dispersion—so long as there is decay to zero, the attentional mass on the sole ‘A’ element in {A B B ... B} will eventually disappear below machine epsilon.
>
> We also remark that Theorem 2.2 shows we cannot get this kind of a situation (a single token decaying with rate $O\left(\frac{\log n}{n}\right)$) in the current way we build frontier Transformer language models, though it might be very interesting to study what kinds of architectures might induce such behaviours!
>
> We continue our response in a new message, due to character limitations.

---

> > ### Author Response · Authors · 2024-11-21
> > **Reply to Reviewer 4NvJ (Part II)**
> >
> > > The statement is too broad and applies equally to any self-attention module in a language model; a more interesting question might be whether self-attention modules in deeper (i.e., later) layers suffer more from this dispersion phenomenon.
> >
> > This is a very nice question, for which we now offer general intuition and a companion special case informal proof (Remark C.1) in Appendix C. The key finding is that, while attention in all layers must disperse eventually (due to Theorem 2.2), there is a sense in which dispersion in earlier layers can exacerbate dispersion in latter layers. This is because, as the attentional coefficients approach a uniform distribution, the output token embeddings get more and more averaged, and this limits the all-important spread factor, $\delta$.
> >
> > In Remark C.1 we informally prove that for BERT-style attention without residual connections, this effect is taken to the extreme: dispersion below macheps in a layer L immediately implies dispersion below macheps in _all_ subsequent layers. We are unable to prove this kind of claim for GPT-style attention or adding residual connections, mainly because such models can never fully average out their token embeddings, and therefore there will always be some $\delta$ remaining, which the MLPs and K/Q matrices can amplify (following Proposition 3.1). That being said, any such counteracting effects cannot last forever, due to Theorem 2.2.
> >
> > ### **Related work on length generalisation and attention sinks**
> >
> > Thank you for calling these out! We now indeed cite length generalisation in our description of the specific out-of-distribution setup we study here:
> >
> > > We define sharp functions by their behaviour as their number of inputs varies. This directly motivates the \emph{out-of-distribution} setting we study: generalising to different amounts of inputs. Specifically, when we analyse neural networks that learn sharp functions, we assume that they are trained on problem instances containing no more than $n$ input items, and we take a particular interest in their sharpness on instances with $n' > n$ items; these are considered \emph{out-of-distribution} instances because they go beyond the maximal number of inputs the model had been prepared for. In language modelling, this setting is also known as \emph{length generalisation} \citep{anil2022exploring}; in graph machine learning, it is known as \emph{size generalisation} \citep{yehudai2021local}.
> >
> > Whereas we note that we had already cited attentional sinks in the original version of the paper -- Darcet et al. who discovered this effect in Vision Transformers.
> >
> > This concludes the core of our response. We hope that it efficiently answers your concerns and improves your view of our work! We are of course very happy to discuss more, if you would like to dive deeper into any of our novel results or call out anything we might have missed, that you consider important at this stage.

---

> > > ### Comment · Reviewer_4NvJ · 2024-11-21
> > >
> > > Thank you for the detailed response.
> > >
> > > **About the presentation**: I briefly reviewed the updated draft and agree that most of the previous issues have been adequately addressed. I have a few minor additional suggestions:
> > >
> > > - I still feel that the statement of Theorem 2.2 could be made more precise to avoid potential difficulties for the reader. This could be achieved by adding a preliminary paragraph and moving some of the relevant notations and definitions there.
> > > - In Eq. (8), the parenthesis after the exponential function is missing.
> > >
> > > **About the theory**: I’m impressed that the authors have produced such fruitful results in such a short time. I’m also glad to see my comments were helpful.
> > >
> > > - The contents in Appendices B and C add substantial value to the paper (much more so than the proofs of Lemma 2.1 and Theorem 2.2). I would suggest moving these proofs to the appendix, and instead highlighting some of the key results in the main text.
> > > - What you wrote in the response makes intuitive sense to me. However, I must admit that I might not have the bandwidth to check the proof of the new results in detail.
> > >
> > > Finally, I agree that the paper is in a much better state, and I am happy to increase my score to 6. That said, I believe (and I expect the authors would agree) that the changes made are substantial. It may be best for the AC/SAC to decide whether another full round of review is warranted.

---

> ### Author Response · Authors · 2024-11-21
> **Thank you!**
>
> Dear Reviewer 4NvJ,
>
> Thank you so much for acknowledging our response and amplifying your score towards acceptance! We highly appreciate that our new theoretical content is seen as useful.
>
> Your remaining presentational suggestions are indeed minor and we are happy to address them in the final revision.
>
> Regarding your remarks on the **theory positioning**:
>
> > The contents in Appendices B and C add substantial value to the paper (much more so than the proofs of Lemma 2.1 and Theorem 2.2). I would suggest moving these proofs to the appendix, and instead highlighting some of the key results in the main text.
>
> We appreciate your suggestion (and are very happy that you feel that the Appendices add valuable content!), though we would like to offer a few points of discussion here:
>
> * We would prefer to keep the proof of Lemma 2.1 in the main body of the paper. This is because it introduces the all-important logit spread quantity, $\delta$, that is used throughout the paper afterwards. It's not easy to motivate why this quantity is important without the proof being there, in our opinion.
>
> * The case for the proof of Theorem 2.2 is more interesting:
>   * It is our opinion that this an important result -- all of the subsequent results we added on the reasoning failures of softmax depend on it being true -- and therefore it is very important that the reader believes us that there are no hidden caveats in the theory.
>   * Also, this is _not a result that feels intuitive at first sight_ (at least, it didn't feel intuitive to us at first).
>
> In comparison, the results in Appendices B and C are indeed important, but we would argue that, once we already know that the coefficients must collapse in Transformers, there is an intuitive sense in which this is going to be bad for the network at some point. At least to us, the results in these Appendices are hence less surprising ("lower entropy") in light of Theorem 2.2 being true, and this inspired the current way we layout the results in the revised paper.
>
> Because of the reasons above, and the constrained space in the main body of the paper, it is our current belief it would do more harm than good to relegate the proof of Theorem 2.2 into appendices. Of course, this is not a belief that cannot be swayed, especially if you feel very strongly about this -- but we believe our case is sensible.
>
> Secondly, on your remark about checking the theoretical work:
>
> > What you wrote in the response makes intuitive sense to me. However, I must admit that I might not have the bandwidth to check the proof of the new results in detail.
>
> We are very happy that the response makes sense, and highly value your time. If you do get some bandwidth in the remaining time to look at the results more, we would suggest focussing on checking the proof of Corollary B.1. Once this result is established, all of the subsequent Remarks either directly follow (as relatively natural specific cases), or require minor analysis on the dataflow of the various Transformer architectures.
>
> Finally, concerning your remark about the changes we made on the whole:
>
> > I believe (and I expect the authors would agree) that the changes made are substantial. It may be best for the AC/SAC to decide whether another full round of review is warranted.
>
> We do agree that the changes are significant in their _volume_, though we also believe that there is a case to be made that they are not as substantial in their _substance_. This is due to the fact that a lot of the modifications we made concern clarifying definitions that already existed in the paper at first (e.g. dispersion, sharpness, OOD), making the paper more self-contained by covering well-known material (e.g. introduction to Transformers), or numerically validating facts we've already proved (e.g. Proposition 3.1.).
>
> It is our desire to be fully transparent about this and invite any debate while we still have a chance to respond to it. As such, later today, we will post a global response (addressed to all Reviewers and the AC) explicitly enumerating the changes made to the paper, along with our opinion on the significance of their substance.
>
> That being said, we are hopeful for your support towards acceptance, given that you found the additional theoretical results to be intuitive and following from Theorem 2.2 (which was already in the paper at the start). We are happy to provide any additional or pointed insight that can resolve any outstanding doubts you have -- we still have plenty of time left.
>
> We sincerely thank you once again for kindly considering our rebuttal and amplifying your score!

---

> ### Comment · Reviewer_4NvJ · 2024-11-21
>
> Thanks again for the response and the summary of the changes.
>
> - I think it is fine to leave the proof of Lemma 2.1 as it is, but I genuinely feel that Theorem 2.2 is a straightforward corollary of Lemma 2.1. Perhaps you could elaborate on why it doesn’t feel intuitive to you? I might be biased (having seen the proof before thinking about the problem from scratch), but to me, the proof is all about "the continuous mapping of a compact set remains compact," which is essentially college-level math.
> - On the other hand, the connection between the dispersion of attention coefficients and the failure of reasoning tasks was not intuitive to me—and still isn’t. Here’s why: although all attention coefficients are on the order of $\frac{1}{n}$, the constants could differ significantly. This is exactly why I brought up the $O(\frac{\log n}{n})$ example. My intuition is that as long as the ground truth token corresponds to a large constant, there is a high chance that the model will function correctly.
> - Finally, I briefly checked the proof of Corollary B.1.
>
>   - The statement: *"At this point, the value of $\alpha_1^{(n)}$ will be indistinguishable from zero, and the weighted sum will reduce to $g(f(v_b))$"* lacks mathematical rigor. The term "indistinguishable" is vague, and it’s unclear what assumptions are being made. To conclude properly, you need to assume that $g$, which maps a vector to a class, is **continuous** at $f(v_b)$. While this is generally true, care must be taken: (1) points on the decision boundary of two classes require special consideration, and (2) this continuity property/assumption should be explicitly stated in the proof.
>   - The example is not very general and even in some sense degenerate. In particular it still does not explain why my previous intuition regarding the large constant might be wrong. I was wondering if this is the best result you can achieve, or do you think it’s sufficient for now and stop exploring?

---

> > ### Author Response · Authors · 2024-11-21
> >
> > Thank you so much for considering our remarks and for taking the time to check our proof!
> >
> > > Perhaps you could elaborate on why it doesn’t feel intuitive to you? I might be biased (having seen the proof before thinking about the problem from scratch), but to me, the proof is all about "the continuous mapping of a compact set remains compact," which is essentially college-level math.
> >
> > Thank you for elaborating on your stance, this prompt was actually quite helpful for us to articulate our view.
> >
> > We are fully in agreement that the _proof technique_ of Theorem 2.2 falls under college-level math and therefore not too interesting -- though, from some of our own personal experiences, we can confirm it is not college-level computer science, so is still likely to be an interesting method for a good chunk of the AI community.
> >
> > To us it is more the very premise that this proof technique can even be applied, and the fact that it is the _tokenised vocabulary_ that is the culprit allowing its application, are the non-intuitive parts. We appreciate that since you already saw and appreciated the proof, it might be hard to convince you of this. But we definitely did not start writing the theoretical section knowing that the tokenisation would be the crucial part in why softmax disperses, or that it would allow us to study continuous mappings between compact spaces.
> >
> > Hopefully our position makes sense!
> >
> > > Here’s why: although all attention coefficients are on the order of $\frac{1}{n}$, the constants could differ significantly.
> >
> > Perhaps this corresponds to a difference in perspective: while it is true the constants can differ, they are still constants, and to us it is intuitive that any constant amplification of mass can be saturated towards a simple average given sufficiently many items with weaker coefficients (that will 'hog the attention' in a cumulative sense). So in this case, it was less surprising to us that such a proof technique has a chance of working.
> >
> > > To conclude properly, you need to assume that $g$, which maps a vector to a class, is continuous at $f(v_b)$.
> >
> > Thank you for catching this!
> >
> > Indeed, the continuity assumption is important assumption which was not explicitly stated in the proof statement -- we apologise for this oversight. We indeed assume $g$ is a continuous function implemented by a feedforward MLP. We have now slightly amended the proof to make the assumption, and where it's being used, clear.
> >
> > > points on the decision boundary of two classes require special consideration
> >
> > Here we assume you are referring to the possible case where the logits output by $g$ could be _literally_ equal for both class $A$ and $B$ on $g(f(\mathbf{v}_b))$, and it would then become a "coin toss" to compute the $\mathrm{argmax}$ logit.
> >
> > We agree this case requires careful consideration, in the sense that we require that such ties need to be broken in a consistent manner (e.g. by choosing the lexicographically smallest logit). Otherwise, we wouldn't be able to ascertain that $\\{A, B, \dots, B\\}$ or $\\{B, \dots, B\\}$ have to be classified identically.
> >
> > We've now added an appropriate sentence and footnote to the end of the proof to make this point clear.
> >
> > Please let us know if we misunderstood you!
> >
> > > The example is not very general and even in some sense degenerate.
> >
> > As we discussed, sharpness implies the capability to isolate a _constant_ amount of items from a _variable-length_ input robustly.
> >
> > This proof demonstrates that isolating a _single_ item from a variable-length input cannot always succeed, no matter how well we prime our softmax architecture. But the rationale extends to retrieving any constant number of items -- the cumulative attention of these items would always be overwhelmed by sufficiently many items being added.
> >
> > In this sense, we believe our example encompasses many instances of sharp retrieval.
> >
> > > I was wondering if this is the best result you can achieve, or do you think it’s sufficient for now and stop exploring?
> >
> > Empirically, softmax-based models will fail to generalise sharp retrieval on many tasks of interest, even at relatively modest sizes (CLRS-Text is one such example across thirty algorithmic tasks).
> >
> > In practice, the classification boundary between our "$A$" and "$B$" classes will often be at some specific $\alpha f(\mathbf{v}_a) + (1 - \alpha)f(\mathbf{v}_b)$, in which case we can show misclassification well before the 'macheps' boundary is hit.
> >
> > But for _proving_ that a particular failure _must_ occur, we cannot rely on empiricism, and we suspect we'd have to introduce assumptions about what classification boundaries are necessary for perfect classification on particular reasoning tasks. Since such boundaries operate over output spaces of MLPs, we would likely need to make sense of these output spaces. This does not sound straightforward given MLPs' universal approximation properties.
> >
> > As such, we indeed believe this is a good place to stop for now. We hope you are in agreement!

---

> > > ### Comment · Reviewer_4NvJ · 2024-11-22
> > >
> > > > Here we assume you are referring to the possible case where the logits output by $g$ could be literally equal for both class
> > >  $A$ and $B$
> > >
> > > This is exactly what I had in mind.
> > >
> > > > We agree this case requires careful consideration, in the sense that we require that such ties need to be broken in a consistent manner (e.g. by choosing the lexicographically smallest logit).
> > >
> > > I believe this is precisely what makes the situation problematic. Recall that continuity is a critical assumption in the proof. Suppose the model predicts $f(v_b)$ as class B. To leverage the dispersion of coefficients, the model needs to consistently predict class B in a small neighborhood of $f(v_b)$. However, if $f(v_b)$ lies on the decision boundary where the logits for both classes are exactly the same, but the model is still specified to classify this point as class B, then the continuity assumption is violated. In this particular case, it seems perfectly fine to me that the model can classify $\alpha_1^{(n)}f(v_a) + (1-\alpha_1^{(n)}) f(v_b)$ as class A for every **non-zero** $\alpha_1^{(n)}$, no matter how small.
> > >
> > > I understand that in practice, the logits for both classes are rarely exactly the same due to floating-point errors. However, from a mathematical perspective, this issue must be formally addressed in the proof.

---

> > > > ### Author Response · Authors · 2024-11-23
> > > >
> > > > Dear Reviewer 4NvJ,
> > > >
> > > > Thank you for your elaboration, this makes perfect sense. Would you be satisfied on the rigour if we add the logit inequality at $f(v_b)$ as an assumption, and remark that this assumption will rarely be violated in practical deep classifiers (for the exact reason you mention)?

---

> > > > > ### Comment · Reviewer_4NvJ · 2024-11-23
> > > > >
> > > > > That sounds like a good plan to me.
> > > > >
> > > > > More broadly, I strongly encourage the authors to carefully review the details of the additional results to ensure there are no missing assumptions or lapses in rigor. Such issues are understandable, given that these results were produced and incorporated into the paper in such a short period of time. I also hope the authors can see, through this example, why I believe that, in an ideal world, another full round of review would be beneficial. That said, based on my interactions with the authors, I am confident that the additional results are reasonably sound and that the paper, in its current form, meets the bar for acceptance.

---

> ### Author Response · Authors · 2024-11-23
>
> Thank you for validating our suggestion, the paper is now revised with the logit inequality assumption explicitly called out.
>
> We highly appreciate the time you have invested in helping us strengthen our theoretical contribution, which we do consider to be among the key takeaways of our work.
>
> And of course we are very happy about your support for acceptance based on our interactions!
> We will make sure to triple-check all of our arguments.

---

### Official Review · Reviewer_CFHF · 2024-10-27

**Soundness:** 3
**Presentation:** 2
**Contribution:** 3
**Rating:** 6
**Confidence:** 3

**Summary:**

The paper aims to show that the widely-used softmax layer must inevitably disperse the output prediction on the out-of-distribution data set. To address this issue, the authors propose an adaptive-temperature technique to improve the sharpness.

**Strengths:**

Overall, I think the paper discusses a quite interesting topic that softmax will basically eventually fail on the out-of-distribution test. The conclusion is indeed supported by some theoretical results and numerical experiments. A partial solution based on adaptive temperature is also provided.

**Weaknesses:**

The theoretical results are not very solid. The presentation needs improvement due to some unclear statements. Adaptive temperature seems only slightly improves the performance.

**Questions:**

- The word ``disperse" appears at the very beginning of the paper. It deserves a precise mathematical definition. I think it should mean that the output distribution will eventually converge to a uniform distribution.

- In Lemma 2.1, the notation $\Theta(\frac{1}{n})$ is used. What is that? I guess it should mean "in the order of $\frac{1}{n}$" according to (4). In fact, the exponential function in (4) quite alerts me. For large $\delta$, these bounds should vary a lot, not behaving like $\Theta(\frac{1}{n})$.

- When considering deep learning's performance on out-of-distribution data, I think it is not surprise at all to see the failure of softmax due to the new statistic information. A more interesting question should be how to quantify it, i.e., the how much change in the output of softmax responds to the change in the distribution of input data.

- Theorem 2.2 seems only consider the worst scenario as it works for networks with any parameters. But this is certainly not true in practice, as those parameters are all trained to minimize the loss functions.

- It seems that Proposition 3.1 can be numerically verified.

---

> ### Author Response · Authors · 2024-11-21
> **Reply to Reviewer CFHF**
>
> Dear Reviewer CFHF,
>
> We would like to thank you for your positive feedback as well as insightful pointers on our theoretical findings. We have now revised our paper in a manner that should address your concerns better, and have follow-up thoughts on some of your points:
>
> ### **Defining dispersion**
>
> Thank you so much for raising this point – we fully agree, and have now concretely specified what we mean by dispersion in the paper:
>
> > Through one of our key theoretical results (Theorem 2.2), we demonstrate that modern deep learning architectures, operating over a fixed vocabulary of input tokens and leveraging the \texttt{softmax} function, are fundamentally incapable of learning functions that remain sharp under such out-of-distribution instances. This is due to the fact that the coefficients emitted by the \texttt{softmax} function must \emph{disperse} as we increase the number of input items. Here by dispersing we mean that, as the number of input items grows, the coefficient attached to each individual item must decay towards zero. This makes it impossible to robustly compute functions that depend on any particular finite amount of input values, such as the aforementioned \texttt{max}, as we show in Appendix B (Corollary B.1 and Remark B.2).
>
> ### **On the use of $\Theta\left(\frac{1}{n}\right)$**
>
> We follow the usual definition of asymptotic growth: a function $f(n)$ is in $\Theta\left(\frac{1}{n}\right)$ if there exist constants $A$ and $B$, such that, as $n\to\infty$, $\frac{A}{n}\leq |f(n)|\leq\frac{B}{n}$.
>
> The dependence on $\delta$ can be abstracted away in the big-Theta notation because $\delta$ is assumed to be a constant – and Theorem 2.2.’s main contribution is, among other things, proving that $\delta$ must be a constant in modern Transformer architectures over tokenised vocabularies.
>
> ### **On the “surprise” of the distribution shift result**
>
> We fully agree with you that it is unsurprising that models tend to behave worse under certain distribution shifts. What _is_ surprising about our result is that our results hold under _any_ statistical distribution of added items, so long as sufficiently many are added. For example, we could add many identical irrelevant items, or a mix of randomly sampled items, and dispersion would still occur in both cases! This, in our opinion, is not a necessarily expected result.
>
> In terms of quantification of how softmax responds to these distribution shifts, we believe that Lemma 2.1. provides a useful quantification of how the dispersive decay rate is controlled, in terms of the spread, $\delta$. We now explicitly call this out in the paper:
>
> > The difference of these bounds (the \emph{spread}, $\delta = \max_{i} e^{(n)}_i - \min_j e^{(n)}_j$) directly controls the rate of dispersion.
>
> ### **On whether Theorem 2.2 considers the worst-case scenario**
>
> From our understanding, Theorem 2.2. does not consider only worst-case scenarios, but actually the _best-case_ scenario.
>
> It proves dispersion will happen out-of-distribution in _every_ self-attention architecture using softmax to attend over all items, _no matter how well it has been trained or which dataset has been used to train it_.
>
> Please let us know if we misunderstood something!
>
> ### **Numerically verifying Proposition 3.1**
>
> Indeed we can do this, and we have! You may find a plot of our upper bound and the empirically observed spread values, over the evolution of a trained model, in Appendix D of the revised paper (Figure 8).
>
> ### **Adaptive temperature only slightly improves the performance**
>
> In our opinion, Figure 7 demonstrates a more significant outperformance of adaptive temperature; namely, it is apparent that the Gemma 2B model gets better at executing several algorithms in CLRS-Text, with gains of 40+% accuracy on several input sizes, when it gains access to adaptive temperature. Since this is the only modification to the architecture, we can attribute these gains to learning to make good use of the adaptation scheme.
>
> Please let us know if there are any other discussion points you have for us during the remainder of the rebuttal period – we are always happy to engage further.

---

### Official Review · Reviewer_kvZv · 2024-10-29

**Soundness:** 3
**Presentation:** 2
**Contribution:** 2
**Rating:** 5
**Confidence:** 3

**Summary:**

This paper addresses the limitations of the softmax function in modern AI. Although softmax is widely used for sharp decision-making in AI, the authors highlight how softmax suffers from dispersion as input data grows. This limits its effectiveness in out-of-distribution scenarios. The authors theoretically establish that softmax inevitably disperses as input grows, preventing sharp focus on specific items in larger datasets. To address dispersion, the authors propose an adaptive temperature mechanism that adjusts softmax sharpness by tuning temperature based on entropy. The authors demonstrate the effectiveness of the proposed method in experiments.

**Strengths:**

* The paper provides a theoretical perspective on the limitations of the softmax function, specifically its inability to maintain sharpness across increasing input sizes.
* The introduction of adaptive temperature as a method to mitigate softmax dispersion is well-motivated.

**Weaknesses:**

* Out of distribution is a key word in this work but the authors do not provide clear definition under the context.
* The performance of proposed method on real-world tasks or datasets is not extensively covered. Additionally, results on only a few benchmarks limit the generalizability of conclusions.
* Although adaptive temperature is a good idea, its implementation could introduce computational tuning complexity, especially in models with many attention heads or large-scale data.
* Not include experiments that assess the scalability of adaptive temperature, especially for high-dimensional, high-volume data.

**Questions:**

In page 1: "Here we call a function sharp if its output only depends on a constant number of its inputs (e.g.max)."
Why does max function only depends on a constant number of its inputs? It is not quit clear here.

---

> ### Author Response · Authors · 2024-11-21
> **Reply to Reviewer kvZv**
>
> Dear Reviewer kvZv,
>
> We would like to thank you for your careful review of our paper, and the ample suggestions for improvement! We hope that our recently-uploaded revision satisfies your concerns.
>
> We also provide more detailed responses to accompany those revisions:
>
> ### **Improving the definition of sharpness**
>
> We fully agree that our previous definition of sharpness did not clearly imply why max is sharp. We have now amended the definition in the paper, with a clear rationale:
>
> > Here we call a function taking a variable number of inputs \emph{sharp} if its output value can be expressed using only a \emph{constant} number of these inputs. For example, \texttt{max} is sharp, as its output value is equal to exactly one of its inputs'. The average function is not sharp, as its output value depends on all of its input values (with factor $1/n$ for each of the $n$ items).
>
> ### **Clear definition of our out-of-distribution setup**
>
> Thank you very much for this suggestion! We wholeheartedly agree, and have now added this paragraph into the paper, drawing directly on our sharpness redefinition:
>
> > We define sharp functions by their behaviour as their number of inputs varies. This directly motivates the \emph{out-of-distribution} setting we study: generalising to different amounts of inputs. Specifically, when we analyse neural networks that learn sharp functions, we assume that they are trained on problem instances containing no more than $n$ input items, and we take a particular interest in their sharpness on instances with $n' > n$ items; these are considered \emph{out-of-distribution} instances because they go beyond the maximal number of inputs the model had been prepared for. In language modelling, this setting is also known as \emph{length generalisation} \citep{anil2022exploring}; in graph machine learning, it is known as \emph{size generalisation} \citep{yehudai2021local}.
>
> ### **On the scalability of adaptive temperature**
>
> Thank you very much for raising this issue! Your remark inspired us to investigate how to most efficiently implement adaptive temperature, and we have successfully developed an iterative streaming algorithm for computing the entropy values on-the-fly, in a manner akin to Flash Attention.
>
> This algorithm allows us to compute adaptive temperature in linear space complexity. Using it, we have successfully scaled adaptive temperature on large context windows – up to 131,072 tokens on a single A100 node. We now fully describe this algorithm in Appendix F. To the best of our knowledge, this algorithm is novel.
>
> ### **On the generalisability of conclusions**
>
> Thank you for your remarks! Since our study concerns out-of-distribution generalisation specifically, we have focused our analysis on tasks requiring out-of-distribution generalisation (such as CLRS-Text, a collection of thirty challenging algorithmic execution tasks across many problem sizes). In most other static benchmarks, it might be very difficult to measure the distribution shift in the test set.
>
> We also remark that focusing on synthetic execution tasks is the standard approach in tasks studying length generalisation in LLMs. As a standard representative we refer to “Exploring Length Generalization in Large Language Models” (Anil et al., NeurIPS’22), which studies only two synthetic problems: parity and variable assignment. In contrast, CLRS-Text studies thirty such problems, with a significant increase in their complexity.
>
> Please let us know if there are any other discussion points you have for us during the remainder of the rebuttal period, or anything we can do to improve your opinion of the paper in the time remaining – we are always happy to engage further.

---

### Official Review · Reviewer_HP5U · 2024-11-04

**Soundness:** 3
**Presentation:** 3
**Contribution:** 3
**Rating:** 8
**Confidence:** 3

**Summary:**

The authors show that the ubiquitous softmax function must provably disperse (i.e. converge to the uniform distribution) as input length increases, so long as the inputs are bounded and the temperature is non-zero. As a result, any softmax-based model where such conditions hold true (e.g. a transformer with a finite vocabulary) cannot approximate sharpness with increasing input lengths. These models therefore cannot generalize out-of-distribution to longer problems, and will fail on tasks where learning sharpness or a low-entropy distribution matter.

In the second part of the paper, the authors propose a fix for the softmax, where the softmax temperature $\theta$ is allowed to vary as a function of input entropy. They (a) demonstrate their approach on a toy problem of max retrieval as well as (b) evaluate it on the CLRS-Text benchmark suite using finetuned Gemma 2B models.

**Strengths:**

1. The paper is, overall, well-written and pleasant to read. The text is lucid and careful, and the diagrams are illustrative. In general, I was able to follow along easily without any confusion.

2. Section 2 is well-constructed and compelling. Even though the conclusions of Lemma 2.1 and Theorem 2.2 are relatively simple and follow directly from compactness of the input features, this work is (to the best of my knowledge) the first to emphasize the link between softmax and sharpness approximation, and consequently, the negative impact on a transformer's ability to generalize to longer problems. These theoretical findings are also backed up by empirical results on a toy dataset. Overall, this is an important observation that is worth highlighting.

3. The authors try to provide a fix in the form of an adaptive temperature parameter, and demonstrate some results on both toy and real-world datasets.

**Weaknesses:**

1. It is a little unclear how significant of a problem the dispersive issue with softmax actually is. As the authors themselves have noted, in various prior work studying transformer mechanisms, the heads appear to be sharp. Whether softmax is "sufficiently" sharp is, after all, dataset and problem dependent. Without a more comprehensive evaluation on real-world datasets, it is hard to tell if the problem is overstated.

2. In general, I think the latter half (Sections 3 and 4) is less compelling:

    - It is not clear from the empirical results that merely having an adaptive temperature parameter is a meaningful fix to counter the dispersive tendencies of softmax. E.g. the results in Table 1 are mostly incremental and the visual differences in Figure 6 are minor.

    - As noted by the authors themselves, the correct adaptive function for $\theta$ is dataset-dependent and determining what this function is can be highly non-trivial in attention models.

    - Related to my first point above, but I think evaluation is a little limited in terms of real-world datasets. It is not entirely clear that adaptive temperature softmax makes a significant difference on a real-world natural-language dataset, e.g. I am curious if it would actually improve on a, say, Q&A task where we want to generalize to answers of different lengths.

**Questions:**

1.  The details of Section 4.2 are a little vague to me. This paragraph:

```
That being said, there is an alternate route to make the Gemma model still benefit from our adaptive
temperature module exactly as-is; it just has to directly learn how to leverage it. As such, in our
CLRS-Text ablation we apply adaptive temperature both during fine-tuning and at inference time.
```

is not entirely clear to me, would the authors be able to explain how exactly adaptive
temperature is implemented here?

---

> ### Author Response · Authors · 2024-11-21
> **Reply to Reviewer HP5U (Part I)**
>
> Dear Reviewer HP5U,
>
> We would like to thank you for generously supporting our submission! Additionally, your review raises important points of discussion, which we now address in a revised paper.
>
> ### **On the significance of dispersion**
>
> We would like to reiterate that the key aspect of our study is out-of-distribution behaviour of softmax coefficients. We now also explicitly call this out in the paper, in the following passage:
>
> > We define sharp functions by their behaviour as their number of inputs varies. This directly motivates the \emph{out-of-distribution} setting we study: generalising to different amounts of inputs. Specifically, when we analyse neural networks that learn sharp functions, we assume that they are trained on problem instances containing no more than $n$ input items, and we take a particular interest in their sharpness on instances with $n' > n$ items; these are considered \emph{out-of-distribution} instances because they go beyond the maximal number of inputs the model had been prepared for. In language modelling, this setting is also known as \emph{length generalisation} \citep{anil2022exploring}; in graph machine learning, it is known as \emph{size generalisation} \citep{yehudai2021local}.
>
> With this in mind, we briefly comment on a few of your observations:
>
> > In various prior work studying transformer mechanisms, the heads appear to be sharp.
>
> In all of these prior works, the heads were studied _in-distribution_ examples (oftentimes, these may have even been in the training data!). We now explicitly call this out in the paper:
>
> > ...and the discovered heads always appear sharp when inspected on in-distribution samples
>
> In conclusion:
>
> > Whether softmax is "sufficiently" sharp is, after all, dataset and problem dependent.
>
> We wholeheartedly agree, though in light of our study, we make explicit that it provably _cannot_ be sufficiently sharp for problems requiring robust behaviour under size distribution shifts.
>
> This is exactly the reason why we chose the CLRS-Text dataset to study the effects of adaptive temperature---it is a collection of thirty challenging algorithmic execution tasks where out-of-distribution generalisation is explicitly required. In most other static benchmarks, it might be very difficult to measure the distribution shift in the test set.
>
> Important real-world scenarios where we often run into distribution shifts include tasks such as scientific discovery and agentic behaviours---both known to be rather challenging for LLMs today, and also really challenging to rigorously evaluate.
>
> ### **On the utility of adaptive temperature**
>
> Your review rightfully realises that adaptive temperature is not meant to be a solution to the problem in Section 2 (which has now been strengthened with additional Corollaries in the Appendices), but rather a diagnostic approach to validate some of the implications of our theory and hopefully stimulate some future research directions.
>
> As such, we do not intend to dwell significantly on these points, though we would like to briefly respond to two of your remarks:
>
> > As noted by the authors themselves, the correct adaptive function for $\theta$ is dataset-dependent and determining what this function is can be highly non-trivial in attention models.
>
> This is true, though our CLRS-Text results leverages the same adaptive temperature scheme we deployed specially for max-retrieval (though with additional fine-tuning), and we believe our results indicate that the model learnt to use such an adaptive temperature just fine given additional training signal. With that in mind:
>
> > It is not clear from the empirical results that merely having an adaptive temperature parameter is a meaningful fix to counter the dispersive tendencies of softmax. E.g. the results in Table 1 are mostly incremental and the visual differences in Figure 6 are minor.
>
> In our opinion, Figure 7 is also an important aspect of these results: it is apparent that the Gemma 2B model gets better at executing several algorithms in CLRS-Text, with gains of 40+% accuracy on several input sizes, when it gains access to adaptive temperature. Since this is the only modification to the architecture, we can attribute these gains to learning to make good use of the adaptation scheme.
>
> We continue our response in a second message due to character limitations.

---

> ### Author Response · Authors · 2024-11-21
> **Reply to Reviewer HP5U (Part II)**
>
> ### **How is adaptive temperature implemented?**
>
> We now elaborate in more detail on how this is done in the paper:
>
> > That being said, there is an alternate route to make the Gemma model still benefit from our adaptive temperature module exactly as-is (i.e., with exactly the same polynomial fit as in Figure 5); it just has to directly \emph{learn} how to leverage it. As such, in our CLRS-Text ablation we apply adaptive temperature both during fine-tuning and at inference time. What this means is, we replace all instances of \texttt{jax.nn.softmax} within all the attentional heads of Gemma 2B with our \texttt{adaptive\_temperature\_softmax} function, both during fine-tuning of the model and during inference. This allows the model to learn how to compute key/query embeddings that can maximally exploit the temperature adaptation.
>
> Additionally, we have been able to deploy the entropy computations in a streaming manner (akin to Flash Attention), allowing us to compute adaptive temperature in linear space complexity, scaling it up to context windows up to 131,072 tokens on a single A100 GPU. The detailed elaboration of this algorithm is provided in Appendix F and, to the best of our knowledge, it is novel.
>
> Please let us know if there are any other discussion points you have for us during the remainder of the rebuttal period – we are always happy to engage further.

---

> ### Comment · Reviewer_HP5U · 2024-12-03
> **Response to Authors**
>
> I thank the authors for their detailed response to my review, as well as to the other reviewers. Overall I accept the comments the authors have made and maintain my score.
>
> As to AC questions, OOD literature is indeed very broad, and my understanding is that this paper focuses on a specific kind of OOD (generalization to increasing problem lengths) and specific model architecture (although as a cursory remark, it does not seem to be too difficult to generalize to other architectures if compactness is the only requirement).
>
> This does limit the scope of the proof and so the impact of the paper. Nevertheless, I think the authors have done a good job tightening up the proofs (including the post-rebuttal edition) and offering a potential solution. More so, I think the paper is an important step in the direction of probing the inductive biases that individual (and often overlooked) modeling choices might make, and how these biases can affect specific kinds of distribution shifts. I think it would be a helpful step towards future research in this direction.

---

> ### Author Response · Authors · 2024-12-03
>
> Dear Reviewer H5PU,
>
> Thank you so much for engaging with our response and accepting our comments. We are delighted that you are maintaining your acceptance score.
>
> Just to note we are unable to see the "AC questions" being referenced here, but the points you raised are worth briefly replying to, in case they are helpful!
>
> We indeed focus on size / length generalisation in this work and focus on the most currently used attentional architectures in our proofs (BERT / GPT / Set Transformers).
>
> While this is of course only one kind of distribution shift, it is a shift that has been very extensively studied in recent years (as evidenced by the vast body of literature on length generalisation in LLMs [1, 2, 3, 4...] or size generalisation in GNNs [5, 6, 7...]).
>
> We agree with the reviewer that, as our proof relies on compactness and continuity, it should easily work with many other attentional architectures relying on softmax. To be fair, already the compositional formulation of the architecture in Theorem 2.2 may be sufficient to describe most such architectures (though it of course most naturally aligns with the Transformer)!
>
> > Nevertheless, I think the authors have done a good job tightening up the proofs (including the post-rebuttal edition) and offering a potential solution. More so, I think the paper is an important step in the direction of probing the inductive biases that individual (and often overlooked) modeling choices might make, and how these biases can affect specific kinds of distribution shifts. I think it would be a helpful step towards future research in this direction.
>
> We are extremely grateful for your assessment, and we fully agree about the future potentials of the work.
>
> Best,
> Authors
>
> [1] Anil _et al._, "Exploring Length Generalization in Large Language Models", NeurIPS'22
>
> [2] Ruoss _et al._, "Randomized Positional Encodings Boost Length Generalization of Transformers", ACL'23
>
> [3] Chowdhury _et al._, "Monotonic Location Attention for Length Generalization", ICML'23
>
> [4] Zhou _et al._, "What Algorithms can Transformers Learn? A Study in Length Generalization", ICLR'24
>
> [5] Bevilacqua _et al._, "Size-Invariant Graph Representations for Graph Classification Extrapolations", ICML'21
>
> [6] Yehudai _et al._, "From Local Structures to Size Generalization in Graph Neural Networks", ICML'22
>
> [7] Buffelli _et al._, "SizeShiftReg: a Regularization Method for Improving Size-Generalization in Graph Neural Networks", NeurIPS'22

---

### Official Review · Reviewer_HYsK · 2024-11-07

**Soundness:** 2
**Presentation:** 3
**Contribution:** 2
**Rating:** 6
**Confidence:** 1

**Summary:**

The paper explores the fundamental limit of the Softmax activation function, which is frequently used to model attentional mechanisms of machine learning, for OODs generalization in reasoning tasks that require sharpness (e.g. finding maxima or second maxima) by exploring a simple _max retrieval_ task. The paper claims that even in that simple task, the networks using Softmax cannot generalize well (length generalization) in those tasks, because it cannot approximate the sharpness with increasing problem size (dispersed property). The paper backs its claim with both theoretical analysis and empirical experiments. Moreover, the paper also proposes a simple method to (somehow) alleviate this dispersed property by proposing an adaptive temperature scaling method.

**Strengths:**

The paper is well-written and easy to follow. The setting and motivation are clear, though it is a bit niche (focus on OODs generalization for tasks that strictly require sharpness). The findings are interesting (though the formal claims have some potential issues, see Weaknesses), and the proposed method is promising.

**Weaknesses:**

First things first, I admit that I am not an expert in this topic. I will leave comments on the novelty and soundness of this paper to other reviewers. Here are some other comments:

## Comments on mathematical notions
There are some potential typos in mathematical notions in this paper, for example:

1. The definition of _sharp function_ in Line 50 is not clear to me. Concretely, I do not think that the $\max$ function only depends on the constant number of its inputs, since it must examine all the inputs to output the maximum number. I think the better statement would be "the $\max$ function output value equal to the value of one of the inputs". But it would definitely break the _sharp function_ definition above.

2. The statement of Theorem 2 and its proof is not rigorous. Here in the statement, the authors say that "$\mathcal{X} \in \mathbb{R}^m$ be an $m$-dimensional input feature _space_", but later say that $|\mathcal{X}| < \infty$. This mathematically means that $\mathcal{X} = \\{\mathbf{0} \\}$, where $\mathbf{0} \in \mathbb{R}^m$. Maybe the authors mean something different (like _set_ instead of _space_?), but they should explicitly state it.

**Questions:**

See weaknesses.

---

> ### Author Response · Authors · 2024-11-21
> **Reply to Reviewer HYsK**
>
> Dear Reviewer HYsK,
>
> We would like to thank you for the support you have expressed towards our paper as well as calling out important points where additional mathematical clarity would be required.
>
> We wholeheartedly agree and have now addressed both of the issues you raised in our revised paper:
>
> ### **On sharpness**
>
> We redefine sharpness in the following way, along with a more clear elaboration on why max is sharp:
>
> > Here we call a function taking a variable number of inputs \emph{sharp} if its output value can be expressed using only a \emph{constant} number of these inputs. For example, \texttt{max} is sharp, as its output value is equal to exactly one of its inputs'. The average function is not sharp, as its output value depends on all of its input values (with factor $1/n$ for each of the $n$ items).
>
> ### **On feature spaces**
>
> Thank you for calling out this issue; we believe that the misunderstanding stemmed from the fact that $\mathcal{X}$ is a _feature space_ rather than a vector space. We have now corrected this to read:
>
> > Let $\mathcal{X}\subset\mathbb{R}^m$ be a set of possible $m$-dimensional input features…
>
> Please let us know if there are any other discussion points you have for us during the remainder of the rebuttal period – we are always happy to engage further.

---

> > ### Comment · Reviewer_HYsK · 2024-11-23
> > **Response to rebuttal**
> >
> > The changes authors made address my concerns. I have no other question and my rating is maintained as is.

---

> > > ### Author Response · Authors · 2024-11-23
> > >
> > > Dear Reviewer HYsK,
> > >
> > > Thank you for acknowledging our response: we are happy that your concerns have been meaningfully addressed and that you are in support of accepting the paper.

---

> > > > ### Comment · Reviewer_HYsK · 2024-11-23
> > > > **Response to the author**
> > > >
> > > > Just to clarify, I am not an expert in this field. I put a rating of 6 but a confidence of 1 (I can only have lukewarm support), and It depends on other reviewers to judge.
> > > >
> > > > But after seeing your rebuttal with other reviewers, I think that it might be better if:
> > > > 1. The authors incorporate the formal definition of what kind of OOD generalization the authors are dealing with. I know it was about length generalization, but I agree with some reviewers that it should have a formal definition before instantiating theoretical results.
> > > >
> > > > 2. If it is about length generalization, might a formal discussion on prior works in this direction help the positioning of this paper?

---

> > > > > ### Author Response · Authors · 2024-11-23
> > > > >
> > > > > Dear Reviewer HYsK,
> > > > >
> > > > > Thank you, we completely understand your position -- we still appreciate your support nonetheless!
> > > > >
> > > > > In response to other Reviewers, we defined the OOD setting as follows in the revision:
> > > > >
> > > > > > We define sharp functions by their behaviour as their number of inputs varies. This directly motivates the \emph{out-of-distribution} setting we study: generalising to different amounts of inputs. Specifically, when we analyse neural networks that learn sharp functions, we assume that they are trained on problem instances containing no more than $n$ input items, and we take a particular interest in their sharpness on instances with $n' > n$ items; these are considered \emph{out-of-distribution} instances because they go beyond the maximal number of inputs the model had been prepared for. In language modelling, this setting is also known as \emph{length generalisation} \citep{anil2022exploring}; in graph machine learning, it is known as \emph{size generalisation} \citep{yehudai2021local}.
> > > > >
> > > > > This is done before any theoretical results are given. As you can see, we also cite Anil et al. and Yehudai et al. that study these kinds of length generalisation in various settings, as well as other empirical papers (e.g. Markeeva et al.) studying length generalisation on algorithmic tasks.
> > > > >
> > > > > Is there anything you'd wish us to modify in this response?
> > > > >
> > > > > Many thanks!

---

> > > > > > ### Comment · Reviewer_HYsK · 2024-11-24
> > > > > > **Response to the author**
> > > > > >
> > > > > > Is there any better definition in the literature? The definition above is not mathematically rigorous to me, and it will potentially hurt your theoretical results (Lemma 2.1. for example, which only holds asymptotically and you cannot really tell what would happen if $\infty > n' > n$). Maybe deriving a finite sample bound could help, but there is not much time left for the author so that request might be unreasonable. However, other reviewers are complaining about the triviality of your theoretical results and their lack of insights (not me, I simply see this paper as an empirical paper and the added results are just the cherry on the cake) so you might want to make it more concrete in that part to avoid being rejected with a good score.

---

> ### Author Response · Authors · 2024-11-24
>
> Dear Reviewer HYsK,
>
> Thank you for elaborating on your position, and we appreciate your suggestions.
> We would also like to do our best to improve the chances of the paper.
>
> > The definition above is not mathematically rigorous to me, and it will potentially hurt your theoretical results (Lemma 2.1. for example, which only holds asymptotically and you cannot really tell what would happen if $\infty > n' > n$).
>
> Could you please elaborate on why Lemma 2.1 cannot tell us what happens for $n' > n$?
>
> The asymptotic behaviour discovered in Lemma 2.1 is obtained through first deriving an established upper and lower bound on the coefficients (Equation 8), which depends on the number of logits, $n$.
>
> Therefore we can use the derivation of Lemma 2.1 to exactly give upper and lower bounds for coefficients for _any_ $n' < \infty$ we want.
>
> In fact, we use this exact bound in our proof of Theorem 2.2, to not just say that all heads must disperse, but give an _exact_ lower bound on $n' < \infty$ after which dispersion below any given $\epsilon > 0$ must happen.
>
> With this in mind, do you have any suggestion for us on how to improve the rigour of the definition?
>
> > However, other reviewers are complaining about the triviality of your theoretical results and their lack of insights (not me, I simply see this paper as an empirical paper and the added results are just the cherry on the cake) so you might want to make it more concrete in that part to avoid being rejected with a good score.
>
> We presume you are referring to Reviewer 4NvJ.
>
> Just to remark, we had a very useful discussion with the Reviewer during the rebuttal period and have managed to provide additional theoretical results, after which the Reviewer improved their score. We are unaware of other Reviewers calling out triviality or lack of insight of the theoretical results (though we haven't yet had a chance to engage with the other Reviewers).
>
> Best,
> Authors

---

### Author Response · Authors · 2024-11-21
**Overview of changes made to the paper (Part I)**

Dear Area Chair, Dear Reviewers,

We would like to thank you all so much for handling our paper and the wonderful insights you've provided us with so far!

As part of our uploaded paper revision, we have made significant improvements to the paper's presentation, clarity, and theoretical evidence -- prompting (so far) one of the Reviewers (4NvJ) to upgrade their score. They also remarked that it is worth taking into account the scale of the changes made to the paper during the rebuttal.

We want to be transparent about the changes we've made, so we tabulate them all in our response, along with our assessment of their level of additional substance compared to the original paper.

It is also our hope that this tabulation will make it easier for every Reviewer to navigate the revised paper efficiently.

### **Our ask:**

If you feel strongly about the scale of these changes, or there is anything you would like to discuss about them, we kindly ask that you please let us know while there is still a chance for us to respond. For example, we may agree to streamlining some of the changes in a further revision, or invite further discussion on the scale of the revisions made.

Thank you all so much once again -- we really appreciate your valuable time!

_[Changes are described in the reply below, due to character limits]_

---

> ### Author Response · Authors · 2024-11-21
> **Overview of changes made to the paper (Part II)**
>
> ### **Clarified definitions of key terms** (Section 1)
>
> * Sharpness
> * Dispersion
> * Out-of-distribution
>
> These are all terms we used and leveraged throughout our original paper. The updated definitions serve to make it very clear (e.g. using mathematical concepts) what these terms mean for the scope of the paper---they _do not_ add any information to the paper that was not at least implied originally.
>
> ### **Primer on Transformers** (Section 1)
>
> We now provide a full introductory material on Transformers, attentional heads and coefficients, prior to using them in our analysis. This constitutes standard knowledge in LLM architecture design, and serves to better introduce the reader to the specific notations we use in the paper. Hence, we _do not_ consider this to be a semantic change to the paper, merely one that improves the ease-of-reading.
>
> ### **Clarifications in various descriptions**:
>
> * Clarifying that the logit spread, $\delta$, controls the constant of the dispersion rate (Section 2)
> * A minor modification to the definition of a feature space using _sets_ rather than _spaces_ (setup of Theorem 2.2)
> * A more precise description of how Gemma 2B experiments leverage adaptive temperature (Section 4.2)
> * Improving descriptiveness of figure captions (Figure 2, 3 and 7)
> * Adding a few recent relevant related work citations across the text
>
> These correspond to minor (often single-sentence level) modifications to the text, and they mainly serve to clarify the description that was already there during the initial submission. We _do not_ consider this to be a semantic change to the paper, mainly a change that improves convenience.
>
> ### **Further theoretical results** (Appendices B and C)
>
> We have derived a useful corollary of Theorem 2.2 (Corollary B.1, Remark B.2, Remark B.3) showing that attention head dispersion in Transformers provably leads to reasoning failures on simple tasks such as max-retrieval. This directly ties our observation to OOD failure modes of language models, and is hence valuable and important.
>
> That said, the line of proof used is relatively straightforward given Theorem 2.2:
> * We know (Thm 2.2) all attention heads' coefficients will eventually disperse towards zero
> * Take a set with one maximal item and $n$ smaller items
> * At some point, the attention coefficient on the max item will be negligible against the total coefficient on other items.
> * Once this coefficient falls below macheps, the set is indistinguishable from a set containing only smaller items, hence those two sets must be classified the same way by a Transformer.
> * This implies at least one of them must be misclassified.
>
> And it should be reasonably intuitive that attention head dispersion can lead to undesirable outcomes. Hence, these corollaries' aim is to validate already established intuition in the original paper, rather than introduce brand-new theoretical advances. As such, we hope they will be seen as _appropriate_ for the rebuttal period.
>
> We also provided an informal analysis of how vulnerable different attention heads are to dispersion as a function of their depth. This mainly relies on an intuitive observation -- dispersion of a given layer will bring embeddings closer to an averaged embedding, limiting embedding (and hence logit) spread, and accelerating dispersion elsewhere. In addition, we informally proved (Remark C.1) that for a specific BERT-style architecture, full dispersion in layer L immediately implies full dispersion in all future layers.
>
> As above, we find these observations to be reasonably intuitive given the results in Theorem 2.2 -- further, the idea of amplifying or attenuating logit spread was already explored in the original paper, by Proposition 3.1 -- and hence we also consider them validation of these intuitions, and hence _appropriate_ for the rebuttal period.
>
> ### **Numerical validation of Prop 3.1** (Appendix D, Figure 8)
>
> We show that Prop 3.1 holds in practice during training of self-attentional models. As we already proved this bound in the original paper, this is just a sanity check and we _do not_ consider it to constitute a substantial change.
>
> ### **Streaming algorithm for attentional entropy** (Appendix F)
>
> This is an algorithm allowing for attentional coefficient entropy to be computed in an _online_ manner, allowing us to serve adaptive temperature in a manner akin to Flash Attention, enabling highly scalable $O(n)$ memory usage for a context window of $n$ tokens.
>
> While this significantly improves the applicability of our adaptive temperature proposal to large language models, the result is derived mainly through careful algebraic manipulation of the individual terms; all steps of the algorithm can be easily checked for correctness. We also numerically verified that the outputs of the online algorithm match (are sufficiently close to) the naïve JAX implementation provided in the original paper.
>
> All things considered, we find this change to be _appropriate_ for the rebuttal.

---

### Meta-Review · Area_Chair_XZfT · 2024-12-21

**Metareview:**

The paper investigates the limitations of the softmax function in ML systems, particularly its inability to maintain sharpness as input size increases, which affects out-of-distribution (OOD) generalization. The title and the introduction focus on OOD, but the definition of OOD in the paper is not clear.  The authors provide theoretical and empirical evidence for this limitation and propose an adaptive temperature mechanism to improve softmax sharpness. This is a difficult case. On one hand, the significance of OOD makes this relevant to the ICLR, but on the other hand, there is a rich literature of OOD papers that is *completely* ignored by the authors. That makes me question the validity of the findings and how those are placed in the literature. For instance, [1] was an oral in ICLR'21 and it should be cited, along with more recent references or related perspectives, e.g., invariant learning. The paper should be improved in terms of writing (most of the reviewers raised concerns about mathematical issues). Also, it would help to clarify whether this issue with the softmax is applicable to all architectures or not, which neither the title nor the abstract are explicit about.

To summarize, this is a paper that could add value in the literature, but there are significant changes to achieve this. Firstly, the writing could be improved and the large literature on the topic should be considered carefully (this is a significant weakness as more recent works, e.g. [2-4], might be relevant depending on the definition of OOD). This is a challenging decision, but I do believe another round of significant improvement and then reviews will be beneficial, especially if I compare with the rest of the accepted submissions in my batch.


[1] How Neural Networks Extrapolate: From Feedforward to Graph Neural Networks, ICLR'21.

[2] Exploring Transformer Extrapolation, AAAI'24.

[3] A Length-Extrapolatable Transformer.

[4] Dissecting Transformer Length Extrapolation via the Lens of Receptive Field Analysis.

**Additional Comments On Reviewer Discussion:**

The reviewers raised several concerns in the original review, including for Theorems being trivial (e.g. Theorem 2 being an immediate corollary of Lemma 1), or that the proposed algorithm does not tackle the issue raised by the authors themselves. In addition, some reviewers raise concerns regarding the real-world applications of this method, which were not addressed during the rebuttal. Overall, most of the reviewers raised concerns regarding mathematical components of this paper, therefore the paper should undergo another round of reviews to ensure the results are rigorously supported.

---

### Decision · Program_Chairs · 2025-01-22

Reject